# Structural analysis of HTL and D14 proteins reveals the basis for ligand selectivity in *Striga*

Yuqun Xu [1], Takuya Miyakawa[1], Shohei Nosaki [1], Akira Nakamura [1], Ying Lyu[1], Hidemitsu Nakamura[1], Umeharu Ohto[2], Hanako Ishida[2], Toshiyuki Shimizu[2], Tadao Asami[1,3] & Masaru Tanokura [1]

HYPOSENSITIVE TO LIGHT (HTL) and DWARF14 (D14) mediate the perception of karrikin and strigolactone, which stimulates germination of the parasitic weed *Striga*. However, their role in parasitic seeds is poorly understood, and the basis for their differing responsiveness remains unclear. Here, we show that *Striga hermonthica* HTL proteins (ShHTLs) in 'conserved' and 'intermediate' clades are able to bind karrikin. The 'divergent' clade is able to hydrolyze strigolactone. Unexpectedly, we find that ShD14 is also capable of hydrolyzing strigolactone. Through comparative analysis of ShHTLs and ShD14 crystal structures, we provide insights into the basis for their selectivity. Moreover, we show that both ShD14 and divergent clade ShHTLs, but not conserved and intermediate clade ShHTLs, can interact with the putative downstream signaling component ShMAX2 in the presence of the synthetic strigolactone, *rac*-GR24. These findings provide insight into how strigolactone is perceived and how ligand specificity is determined.

[1] Department of Applied Biological Chemistry, Graduate School of Agricultural and Life Sciences, The University of Tokyo, 1–1–1 Yayoi, Bunkyo-ku, Tokyo 113–8657, Japan. [2] Graduate School of Pharmaceutical Sciences, The University of Tokyo, 7–3–1 Hongo, Bunkyo-ku, Tokyo 113–0033, Japan. [3] Department of Biochemistry, King Abdulaziz University, Jeddah 21589, Saudi Arabia. Correspondence and requests for materials should be addressed to M.T. (email: amtanok@mail.ecc.u-tokyo.ac.jp)

Strigolactones (SLs) and karrikins (KARs) are two classes of structurally related molecules that mediate signaling pathways to stimulate seed germination. SLs induce seed germination of the *Striga* genus[1], which are obligate hemiparasites that parasitize the roots of important crops in sub-Saharan Africa[2], thus causing a reduction in agricultural production by 30–90%[3,4]. In particular, *Striga hermonthica* causes more crop loss than any other individual parasite in Africa. In addition to germination stimulation, the root-secreted SLs in the soil also mediate symbiotic associations with arbuscular mycorrhizal fungi[5], and SLs are also endogenous phytohormones that regulate several aspects of plant growth, such as shoot branching[6,7]. KARs are post-fire germination stimulants found in the smoke of burning vegetation[8,9] and have been found to regulate seedling development in *Arabidopsis*[10,11]. Although KARs trigger the germination of numerous plant species, they fail to induce the germination of parasitic weeds[12,13].

SL and KAR have been reported to be recognized by the homologous proteins D14 (DWARF14)[14–17] and HTL (HYPO-SENSITIVE TO LIGHT; also known as KAI2, KARRIKIN INSENSITIVE2)[18–20], respectively. D14 requires the F-box protein MAX2 (MORE AXILLARY GROWTH2) as the SL signal-transducing component, and MAX2 is also required for KAR signaling[21–25]. It has been reported that D14 is attacked by SL to produce a covalently linked intermediate molecule (CLIM), which induces a conformational change in D14 to facilitate an interaction with MAX2 to trigger SL signaling[17]. By contrast, less is known about KAR signaling. It has been suggested that MAX2 and HTL/KAI2 are involved in KAR signaling based on the experimental observation of the KAR insensitivity of *max2* and *kai2-2* mutants[18,23]. The KAR-induced and *rac*-GR24-induced AtHTL–AtMAX2 interaction has been reported using yeast two-hybrid (Y2H) assays[26,27].

Although signals of endogenous SL as phytohormones are transduced via the D14–MAX2 system, the perception of exogenous SL in parasite seeds is not well understood. Recently, the perception of SL by *S. hermonthica* has been reported to be mediated by *HTL* homologs[13,28–30]. Eleven *S. hermonthica* HTLs (ShHTLs) have been identified and categorized into three phylogenetic clades: a conserved clade (*ShHTL1*, also referred to as *KAI2c*), an intermediate clade (*ShHTL2* and *ShHTL3*, also referred to as *KAI2i*), and a divergent clade (*ShHTL4–11*, also referred to as *KAI2d*) (Supplementary Fig. 1). The conserved clade is under the strongest purifying selection and found in most plants, while the divergent clade is fast evolving and parasite specific. ShHTLs of the different clades showed distinct responses to SL, including natural SLs: strigol, 5-deoxystrigol (5DS), 4-deoxyorobanchol and sorgolactone, and a racemic mixture of the synthetic SL analog GR24 (*rac*-GR24) or KAR in cross-species complementation assays, in which *Arabidopsis kai2-2* and *htl-3* mutants were transformed with *ShHTLs* genes: *ShHTL1* is not responsive to KAR or SL; *ShHTL2* and *ShHTL3* are responsive to KAR but not to SL; and *ShHTL4–9* are responsive to SL but not KAR[13,29]. Despite these ShHTLs proteins having a high sequence conservation (sequence identity of more than 60%), they have distinct SL/KAR specificities and are therefore important for the study of SL/KAR perception. Unveiling the structural basis for the ligand specificities of this series of proteins will help us understand the evolution and discrimination of SL and KAR signals. In contrast, although only one D14 ortholog, ShD14, has been found in *Striga*, it showed no enzymatic activity toward Yoshimulactone Green (YLG), a fluorogenic SL agonist that becomes fluorescent when it is hydrolyzed by AtD14 or some HTL enzymes[27,28], despite possessing conserved catalytic triad residues (Supplementary Fig. 2). The molecular mechanism underlying the different responsiveness of these ShHTLs and ShD14 remains vague.

In this study, we evaluate the SL and KAR recognition ability of ShHTLs and ShD14 in vitro and determine the crystal structures of different clades of ShHTLs and ShD14, providing structural insights into the evolution of their SL/KAR selectivity based on a comprehensive structural comparison. Furthermore, interactions between ShHTLs and the putative downstream signaling component ShMAX2 (*S. hermonthica* MAX2) are determined comprehensively in the present study. These results reveal structural insights into the evolution of ligand specificity of ShHTLs and ShD14 and will provide information for the design of *Striga* germination stimulants.

## Results

**Ligand specificities of ShHTLs and ShD14.** The SL and/or KAR responsiveness of ShHTLs has been characterized using transgenic *Arabidopsis* lines transformed with *ShHTL* genes in the *kai2-2* or *htl-3* mutant background[13,29]. However, the signaling machinery is derived from *Arabidopsis*, and it is still unknown whether the ShHTL homologs bind certain ligands *in planta*. Based on our previous report[31] on the ligand specificity of the intermediate clade HTL (ShHTL3, also referred to as ShKAI2i and sharing 98% identity with ShKAI2iB), we evaluated the ligand specificities of the conserved clade HTL (ShHTL1), the divergent clade HTLs (ShHTL4 and ShHTL7), and ShD14. Their KAR$_1$ binding abilities were determined using isothermal titration calorimetry (ITC) assays (Fig. 1a and Supplementary Fig. 3a−d). Only ShHTL1, in the conserved clade, was capable of binding KAR$_1$ (dissociation constant ($K_D$): 77 ± 12 μM; means ± standard deviation (SD), $n = 3$) with an affinity comparable to that of ShHTL3 of the intermediate clade ($K_D$: 78 ± 3 μM)[31]. AtHTL is known to be required for KAR responses in *Arabidopsis*[18,23]. Its binding affinity to KAR$_1$ was also measured using ITC assays in the present study, and the $K_D$ value was determined to be 129 ± 17 μM (Supplementary Fig. 3e), which was comparable to ShHTL1 and ShHTL3. Our results for ShHTL1 binding KAR$_1$ were unexpected because it has been reported that ShHTL1 is not responsive to either KAR$_1$ or *rac*-GR24 in cross-species complementation assays in terms of seed germination, but rescues some other phenotypes[13,32] (discussed later). Moreover, no significant KAR$_1$ binding ability of ShHTL4, ShHTL7, and ShD14 was observed in the present study (Fig. 1a; Supplementary Fig. 3b−d).

The SL hydrolytic activity of ShHTLs and ShD14 is important for their function as SL receptors. Therefore, hydrolytic activities toward synthetic SL *rac*-GR24 and natural SL 5DS were measured by detecting the amount of SL by high-performance liquid chromatography (HPLC) after enzyme treatment (Fig. 1b). ShHTL4 and ShHTL7 were able to hydrolyze both *rac*-GR24 and 5DS, consistent with a previous report[28] in which YLG was used to measure their enzymatic activity and their responsiveness to *rac*-GR24 and 5DS in cross-species complementary assays[13,29]. Here, we also demonstrated the hydrolytic activity of ShD14 toward *rac*-GR24 and 5DS, despite the demonstration of no hydrolytic activity toward YLG in a previous study[28]. To further confirm these results, we repeated the YLG hydrolysis assays using ShD14 and OsD14 and obtained the same results as previously published[28] (Supplementary Fig. 4). ShD14 was able to hydrolyze *rac*-GR24 and 5DS but not YLG, suggesting that ShD14 might also serve as an SL receptor. In contrast, ShHTL1 and ShHTL3 showed no hydrolytic activity toward *rac*-GR24, while ShHTL1 had weak 5DS hydrolytic activity, which was so weak that 5DS-dependent germination was not observed in transgenic lines[29]. Therefore, our results suggested that ShHTL4, ShHTL7, and ShD14 were able to hydrolyze *rac*-GR24 and 5DS, but lacked KAR$_1$ binding ability. In contrast, ShHTL1 and ShHTL3 were KAR$_1$ binding proteins without hydrolytic activity toward *rac*-GR24.

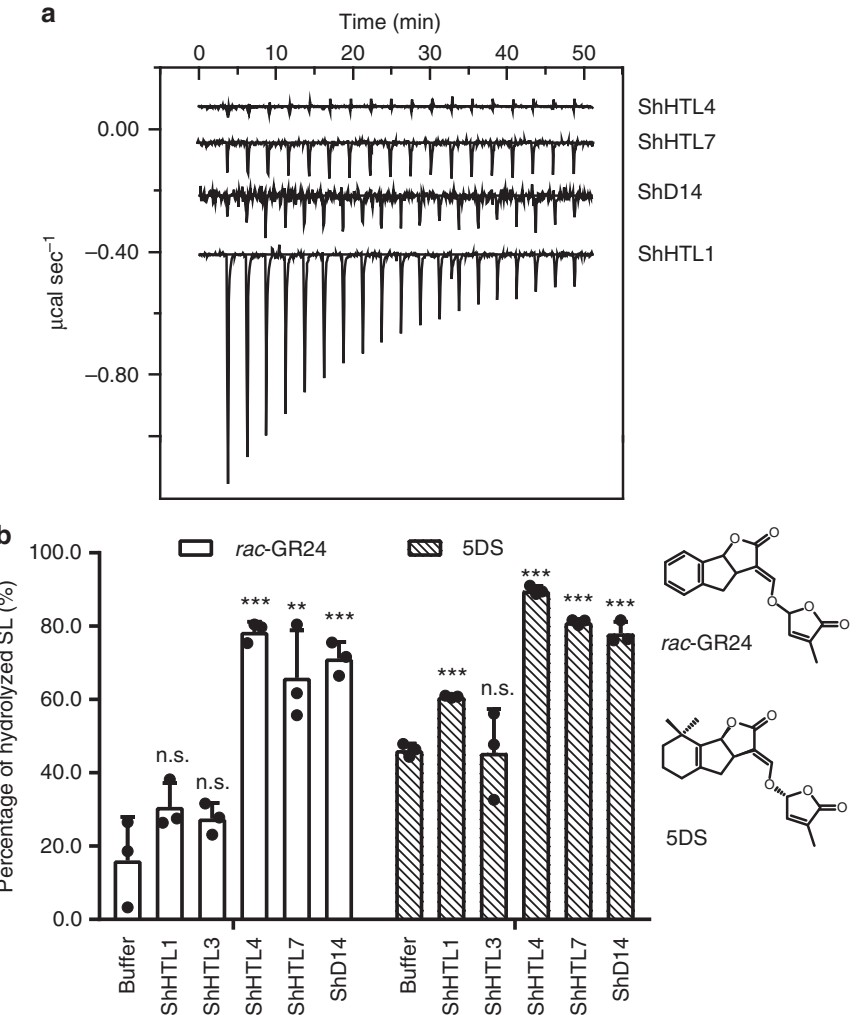

**Fig. 1** In vitro ligand specificity of ShHTLs and ShD14. **a** ITC thermograms titrating KAR$_1$ into ShHTL1, ShHTL4, ShHTL7, and ShD14, respectively. $K_D$ of KAR$_1$ to ShHTL1 was 77 ± 12 μM, calculating from three independent experiments (means ± standard deviation (SD), $n = 3$). **b** Hydrolysis of *rac*-GR24 and 5DS by ShHTLs. *rac*-GR24 or 5DS was incubated with or without purified enzymes (ShHTL1, ShHTL3, ShHTL4, ShHTL7, and ShD14) for 30 min at 30 °C. HPLC analysis was used to detect the remaining amount of *rac*-GR24 and 5DS. Columns represent the mean value and error bars the SD of three replicates (means ± SD, $n = 3$). Each replicate is indicated by a black dot. The asterisks indicate statistical significance from the buffer (non-enzyme sample) as ***$p \leq 0.001$; **$p \leq 0.01$ and n.s., $p > 0.05$, as measured by the one-tailed unpaired $t$- test

**Structural differences in the helix αD1**. To elucidate the structural basis of ligand specificity, we determined the crystal structures of ShHTL1, ShHTL4, ShHTL7, and ShD14 (Fig. 2, Supplementary Fig. 5 and Table 1) and compared their structures with those reported for ShHTL3[31] and ShHTL5[29]. All the structures were highly similar to one another, consisting of a canonical α/β hydrolase domain and a helical cap. However, there were apparent structural differences in the helical cap, especially in helices αD1 (maximum shift of approximately 5 Å) and αD2 (Fig. 3a, b). This flexibility of the cap domain is consistent with the observation that helices αD1, αD2, and αD3 in D14 underwent a large structural change upon binding to MAX2[17]. The structural differences in the cap domain appeared to arise from the Y150/F150 residue on loop αD1–αD2 (Fig. 3c). Y150 was changed to F150 in a subgroup (ShHTL4, ShHTL5, and ShHTL7) of the divergent clade, thus losing the hydrogen bond between helices αD1 and αD3 (Fig. 3d). Therefore, helices αD1 and αD2 of ShHTL4, ShHTL5, and ShHTL7 slid away from αD3, resulting in larger pockets with a reduced chance of causing steric hindrance with ligands. This finding is interesting because ShHTL4,

ShHTL5, and ShHTL7 have been shown to be highly sensitive to various natural SLs (polyspecific receptors) in a previous study[29]. Therefore, it is speculated that owing to the change in residue 150, *Striga* can sense the existence of hosts at very low SL concentrations and recognize various types of SLs that are secreted from hosts.

**Structural basis for different ligand specificities**. It has been proposed that the pocket size may be the determining factor for recognizing either KAR or SL[33] because KARs are smaller molecules than SLs or GR24. The pocket volume of ShHTLs and ShD14 was estimated to investigate the relationship between the ligand specificity and the pocket size (Fig. 4a). Although similar pocket sizes of ShHTL1 and ShHTL3 have been predicted by modeling, our data showed that the ligand-binding pocket of ShHTL3[31] was smaller than that of ShHTL1. Such a structural difference between ShHTL1 and ShHTL3 might result in distinct functions, such as the different responses in the germination assays[13,29]. Since structural differences were observed in helices αD1, and helices αD1 and αD2 compose the entrance of the

ligand-binding pocket, they might contribute to the different sizes of the ligand-binding pockets of ShHTLs and ShD14. It is apparent that the more the helix αD1 shifts outward (Fig. 3b), the larger the ligand-binding pocket becomes (Fig. 4a). For example, ShHTL7, with the largest pocket, had the helix αD1 tilting

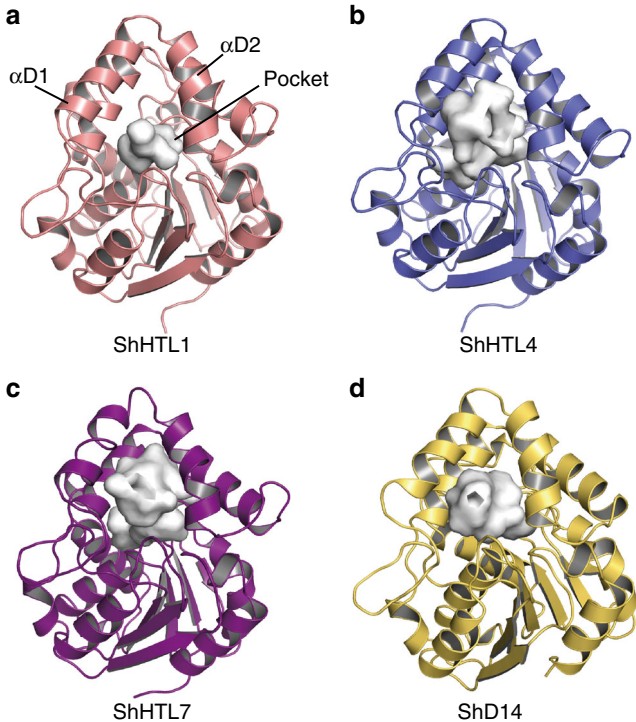

**Fig. 2** Overall structures of ShHTLs and ShD14. **a–d**Structures of ShHTL1 (**a**), ShHTL4 (**b**), ShHTL7 (**c**), and ShD14 (**d**) are colored pink, slate blue, purple, and yellow, respectively. The ligand-binding pockets are shown in surface representations in white

the greatest extent away from the entrance of the pocket. Since the helix αD1 constituted a part of the entrance to the ligand-binding pocket, it is suggested that the orientation of helix αD1 affected the size of the ligand-binding pockets, and thus the helix αD1 might be an important structural determinant for the ligand- binding specificity.

ShHTLs of the divergent clade had larger pockets than the D14 group (Fig. 4a). In the divergent clade (ShHTL4, ShHTL5, and ShHTL7), ShHTL7 had the largest pocket. Structural analyses revealed that although the cap domains of ShHTL7 could overlay well with those of ShHTL4 and ShHTL5, with a root-mean-square deviation (RMSD) of 0.9 Å (Supplementary Fig. 6), ShHTL7 had more non-bulky residues than ShHTL4 and ShHTL5, such as T142, L153, and T157 (Fig. 4a, b). It is reasoned that ShHTL7 evolved to use smaller residues to enlarge its pocket while sustaining activity. The larger pocket of ShHTL7 might be related to its high sensitivity to *rac*-GR24 and its responses to various SLs[28,29] due to a reduced chance of steric hindrance upon SL binding. ShD14 and OsD14 have bulky residues in their pockets, such as F125$^{ShD14}$/F176$^{OsD14}$, W154$^{ShD14}$/W205$^{OsD14}$, and F194$^{ShD14}$/F245$^{OsD14}$ (corresponding to residues 124, 153, and 194 in ShHTLs), making their pockets smaller than those of the divergent clade (Fig. 4a, c). However, these differences between ShD14 and ShHTL7 did not seem to influence their hydrolytic activity toward *rac*-GR24 (Fig. 1b).

The KAR-binding HTLs (ShHTL1, ShHTL3, and AtHTL) had smaller pockets than the D14 group and ShHTLs of the divergent clade due to more bulky and hydrophobic residues. Especially, residue 190 was substituted by a larger side-chain residue in ShHTLs of the conserved and intermediate clades (L190$^{ShHTL1}$/F190$^{ShHTL3}$) (Fig. 4a, d). ShHTL1 and ShHTL3 can bind to KAR$_1$ but not *rac*-GR24 (Supplementary Fig. 7)[31]. The ligand-binding pocket of ShHTL1 (321 Å$^3$) was composed of the largest number of bulky residues (Fig. 4a), resulting in a smaller pocket than AtHTL (411 Å$^3$). The larger pocket size of AtHTL might be the reason why AtHTL exhibited a slightly weaker affinity to KAR$_1$ than ShHTL1 and was able to hydrolyze *rac*-GR24 to some

**Table 1 X-ray data collection and refinement statistics**

|  | ShHTL1 | ShHTL4 | ShHTL7 | ShD14 |
|---|---|---|---|---|
| *Data collection* |  |  |  |  |
| Space group | $P2_12_12$ | $C222_1$ | $P1$ | $P2_1$ |
| Cell dimensions |  |  |  |  |
| *a, b, c* (Å) | 81.2, 138.8, 49.4 | 63.8, 78.0, 99.0 | 46.1, 72.6, 81.1 | 44.7, 70.3, 84.7 |
| α, β, γ(°) | 90.0, 90.0, 90.0 | 90.0, 90.0, 90.0 | 97.7, 95.4, 107.1 | 90.0, 94.0, 90.0 |
| Resolution (Å) | 50.0−1.66 (1.70−1.66)$^a$ | 50.0−2.06 (2.11−2.06) | 50.0-1.90 (2.01-1.90) | 50.0−1.98 (2.03−1.98) |
| Redundancy | 6.7 (6.2) | 3.6 (2.3) | 2.3 (2.2) | 3.8 (3.8) |
| Completeness (%) | 99.4 (93.7) | 98.9 (86.8) | 93.8 (87.0) | 99.6 (99.8) |
| $R_{sym}$ (%) | 6.6 (71.2) | 4.2 (28.8) | 9.6 (37.4) | 5.9 (49.5) |
| < $I/\sigma(I)$ > | 18.0 (2.2) | 18.6 (2.5) | 7.9 (2.2) | 17.6 (3.0) |
| *Refinement* |  |  |  |  |
| Resolution (Å) | 39.0−1.66 | 39.0−2.06 | 43.6-1.90 | 38.3−1.98 |
| No. reflections | 127,260 | 29,369 | 72,939 | 36,501 |
| $R_{factor}$/$R_{free}$ (%) | 16.9/19.9 | 18.9/23.0 | 21.1/25.3 | 20.8/25.4 |
| No. atoms |  |  |  |  |
| Protein | 4199 | 2091 | 8337 | 4105 |
| Water | 819 | 189 | 712 | 459 |
| *B-factors* (Å$^2$) |  |  |  |  |
| Protein | 24.1 | 36.2 | 25.7 | 28.0 |
| Water | 38.0 | 42.4 | 30.1 | 33.1 |
| R.m.s. deviations |  |  |  |  |
| Bond lengths (Å) | 0.004 | 0.002 | 0.004 | 0.004 |
| Bond angles (°) | 0.657 | 0.549 | 0.663 | 0.648 |

$^a$Values in parentheses are for the highest resolution shell

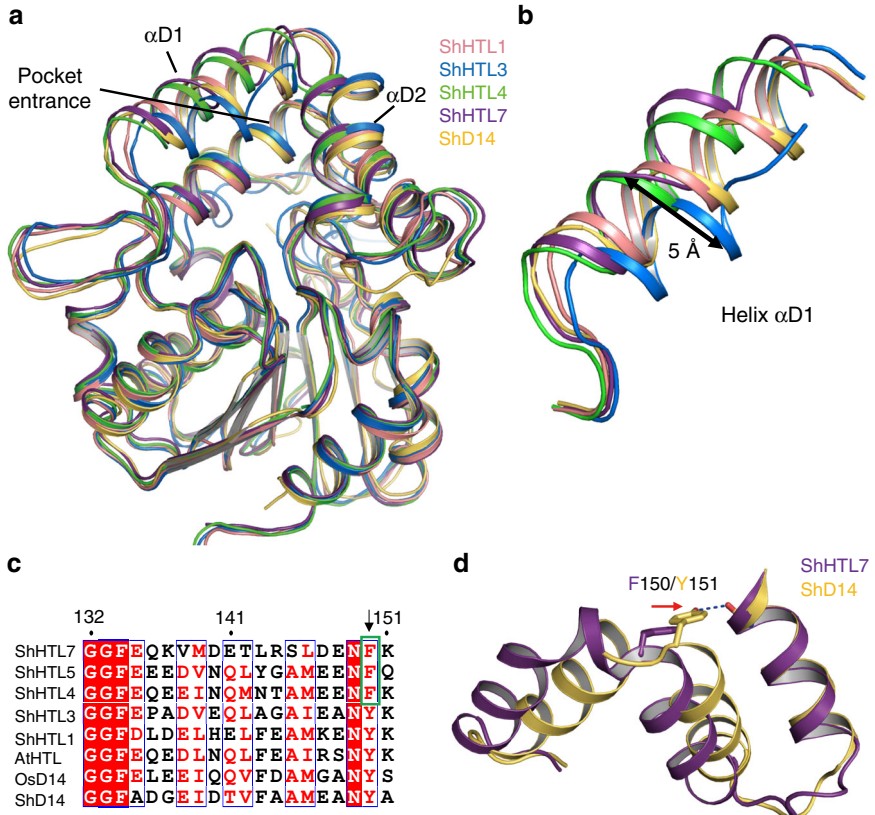

**Fig. 3** Structural comparison of ShHTLs and ShD14. **a** Structural overlay of ShHTL1, ShHTL3 (PDB code: 5DNW [https://doi.org/10.2210/pdb5DNW/pdb]), ShHTL4, ShHTL7, and ShD14 are colored in pink, blue, green, deep purple, and yellow, respectively. **b** A detailed view of helix αD1 is shown. The maximum difference between ShHTL3 and ShHTL7 (approx. 5 Å) is shown by an arrow. **c** Sequence alignment of residues 132−151 of HTLs and D14s. The green box shows the F150 of ShHTLs in the highly sensitive subgroup (ShHTL4, ShHTL5, and ShHTL7). **d** Structural alignment of ShHTL7 (deep purple) and ShD14 (yellow) with the hydrogen bond between Y151$^{ShD14}$ and L178 (main chain carbonyl oxygen atom) is shown as a dashed line

extent[26,34] and hydrolyze the synthesized desmethyl-YLG (dYLG), which is a smaller version of YLG[27]. Although residues 139 and 143 of ShHTL3 were smaller than those of ShHTL1, ShHTL3 had a shifted helix αD1 and a closed entrance to the ligand-binding cavity[31], resulting in the smallest pocket. Due to major bulky residues and the arrangement of the helix αD1, the pocket sizes of ShHTL1 and ShHTL3 were restricted so that there was only enough space to accommodate KAR.

To verify the importance of pocket size for ligand recognition of ShHTLs, a double mutation T190F/C194F and a triple mutation L124F/T190F/C194F of ShHTL7 were generated to introduce large side-chain amino acids to reduce the pocket size by partially mimicking ShHTL3. Both ShHTL7 mutants were able to bind to KAR$_1$ weakly (Supplementary Fig. 8). The inhibition constant $K_i^{rac\text{-}GR24}$ in the YLG competition assays was used to indicate the *rac*-GR24 binding affinity of ShHTL7. It is shown that the T190F/C194F mutation led to a twofold decrease, and the L124F/T190F/C194F mutation led to more than a 100-fold decrease in the *rac*-GR24 binding affinity (Supplementary Fig. 8). Mutating the residues of the ligand-binding pocket into larger residues reduced *rac*-GR24 binding and hydrolyzing ability. These results suggested that the bulky residues in the ligand-binding pocket are required to determine the ligand selectivity of ShHTLs.

A structural overlay with the reported GR24$^{5DS}$-bound OsD14 structure (PDB code: 5DJ5 [https://doi.org/10.2210/pdb5DJ5/pdb]) was performed to investigate other determinants of SL selectivity. The structural alignment revealed that GR24$^{5DS}$ might cause steric hindrance with L142, L/F190, and F194 of the

KAR-binding proteins ShHTL1, ShHTL3, and AtHTL (Supplementary Fig. 9). In contrast, the corresponding residue of L142 in the D14 group was the highly conserved V143$^{ShD14}$ (Supplementary Figs. 9d and 10), which avoids such steric hindrance. Notably, AtHTL hydrolyzes non-natural SLs with a 2'S configuration, such as GR24$^{ent-5DS}$, instead of GR24$^{5DS}$, which is of natural configuration similar to 5DS[34]. A previous study has also shown that the non-natural *ent*-5DS and GR24$^{ent-5DS}$ are able to induce responses similar to SL responses, such as inhibition of hypocotyl elongation and germination of *Arabidopsis* through AtHTL instead of AtD14[35]. GR24$^{ent-5DS}$, with an ABC ring in an opposite orientation from GR24$^{5DS}$, likely avoiding the steric clash derived from L142 (Supplementary Fig. 11) and allowing AtHTL to recognize only GR24$^{ent-5DS}$. Moreover, residue 142 of the divergent ShHTLs did not clash with GR24$^{5DS}$ due to the outward shift of helix αD1. Similarly, the overlay of the KAR$_1$-bound AtHTL (PDB code: 4JYM https://doi.org/10.2210/pdb4JYM/pdb) and ShHTL3 (PDB code: 5DNU https://doi.org/10.2210/pdb5DNU/pdb) structures with GR24-binding proteins revealed that residues 157, 218, and/or 219 of the divergent clade ShHTLs and D14 sterically clashed with KAR$_1$ (Supplementary Fig. 12). Therefore, in addition to an appropriate pocket size, residues 142, 157, 218, and 219 are suggested to be determinants of KAR/GR24 selectivity.

**Interactions with downstream components**. Since the ligand specificity of HTL and D14 is highly related to the signal transduction of these ligands, it is necessary to investigate the

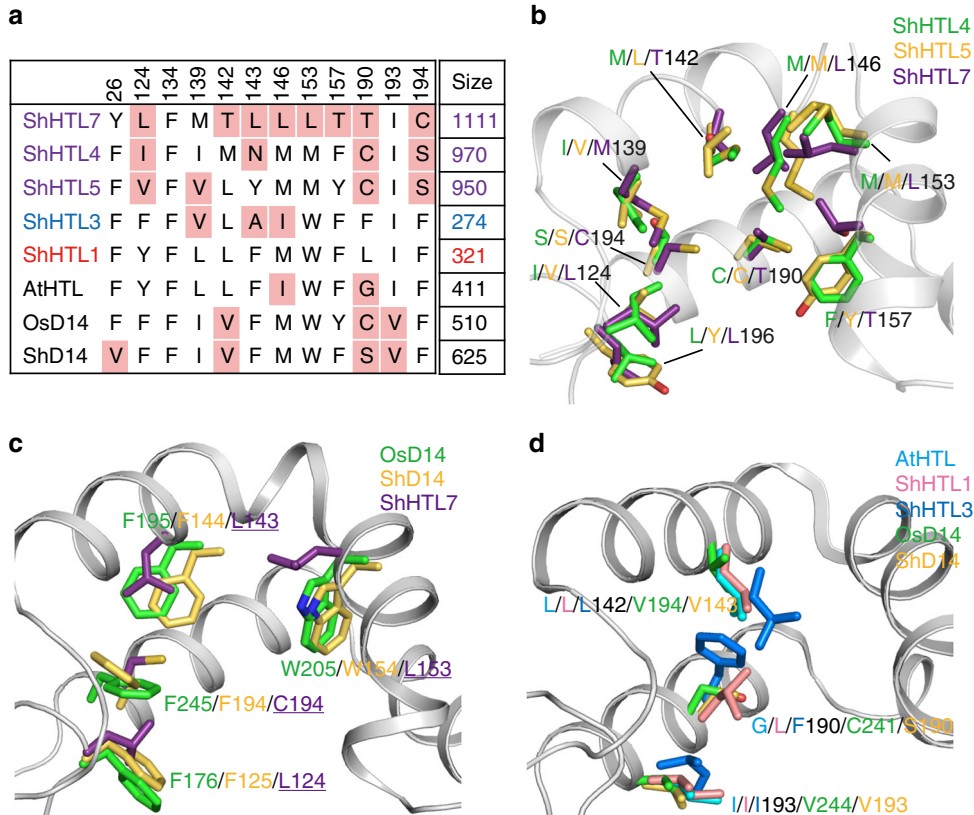

**Fig. 4** Residues involved in determining the pocket size of ShHTLs and ShD14. **a** Residues composing the ligand-binding pocket of each HTL and D14. Font colors of purple, blue, and red represent divergent, intermediate, and conserved clades, respectively, and the background pink color represents less bulky residues. The top row represents the residue number of ShHTL7, and the right column represents the pocket size (Å³). **b** Comparison of residues of ShHTLs of the divergent clade. Residues in the ligand-binding pockets of ShHTL4 (green), ShHTL5 (PDB code: 5CBK [https://doi.org/10.2210/pdb5CBK/pdb]; yellow), and ShHTL7 (purple) are shown. **c** Residues composing the ligand-binding pockets of OsD14 (PDB code: 3W04 [https://doi.org/10.2210/pdb3W04/pdb]; green), ShD14 (yellow), and ShHTL7 (purple) are shown. **d** Residues composing the ligand-binding pockets of AtHTL (PDB code: 4JYP [https://doi.org/10.2210/pdb4JYP/pdb]; cyan), ShHTL1 (pink), ShHTL3 (blue), OsD14 (PDB code: 3W04 [https://doi.org/10.2210/pdb3W04/pdb]; green), and ShD14 (yellow) are shown

ligand-dependent interactions of ShHTLs and ShD14 with their downstream partners. We first investigated the interactions of ShHTLs and ShD14 with ShMAX2 by using Y2H assays (Fig. 5a). Our results showed that *rac*-GR24 triggered the interaction of ShHTL7 with ShMAX2, which was consistent with the results of a previous study[30]. A *rac*-GR24-dependent ShD14–ShMAX2 interaction was also observed, which strongly suggested that ShD14 might serve as an SL receptor. ShD14 showed a high sequence conservation with other angiosperms D14 (e.g., sequence identity of 75% with AtD14 and OsD14) and maintained one *D14* copy in *Striga* as in other nonparasitic plants[13], which strongly suggested that ShD14 had a conserved function with D14 from other angiosperms. In contrast, *AtD14* failed to rescue the germination ability of *kai2-2* mutants[13], suggesting that D14 was not involved in germination. Moreover, the transcripts of ShD14 remained unchanged during the conditioning and germination process of *Striga* seeds;[28] therefore, it was likely that ShD14 did not function in germination. Instead, ShD14 might perceive endogenous SL as a phytohormone in *Striga*, similar to other D14 proteins in nonparasitic plants[17], which would be consistent with the existence of endogenous SL synthesis components in *Striga*[36]. Unexpectedly, KAR-binding ShHTL3 failed to interact with ShMAX2 in the presence of KAR₁. Both of ShHTL1 and AtHTL interacted with ShMAX2 nonspecifically in a ligand-independent manner.

Since Y2H assays readily detect very weak interactions, in vitro pull-down assays were also performed to further examine the interactions with MAX2. It has been reported that AtMAX2 interacts with AtD14 in a *rac*-GR24-dependent manner;[17] therefore, the interaction between AtMAX2 and AtD14 was detected as a positive control (Fig. 5b). Consistent with the results of the Y2H assays, a direct interaction was detected between ShMAX2 and ShD14 in a *rac*-GR24-dependent manner, as in the AtD14–AtMAX2 interaction. ShMAX2 was also able to interact with ShHTL4 of the divergent clade in the presence of *rac*-GR24 (Fig. 5c). No obvious interaction between ShHTL3 and ShMAX2 was detected, but a nonspecific interaction of ShHTL3 with the resin was detected. ShHTL3 was responsive to KAR₁ in *Arabidopsis*[13] and able to bind to KAR₁ in vitro[31], but the involvement of ShHTL3 and MAX2 in the KAR signaling pathway remains an enigma. Finally, neither the ShHTL1–ShMAX2 (Fig. 5c) nor AtHTL–AtMAX2 (Fig. 5b) interactions were detected. Since weak interactions were detected using Y2H assays, western blotting was performed to further examine this phenomenon. A weak ligand-independent ShHTL1–ShMAX2 interaction was detected by western blotting, while the AtHTL–AtMAX2 interaction was enhanced by *rac*-GR24 (Supplementary Fig. 13a). These interactions were further confirmed using the Y2H system with a dilution series (Supplementary Fig. 13b, c). A significant difference between the control and the presence of KAR₁ or *rac*-GR24 was not

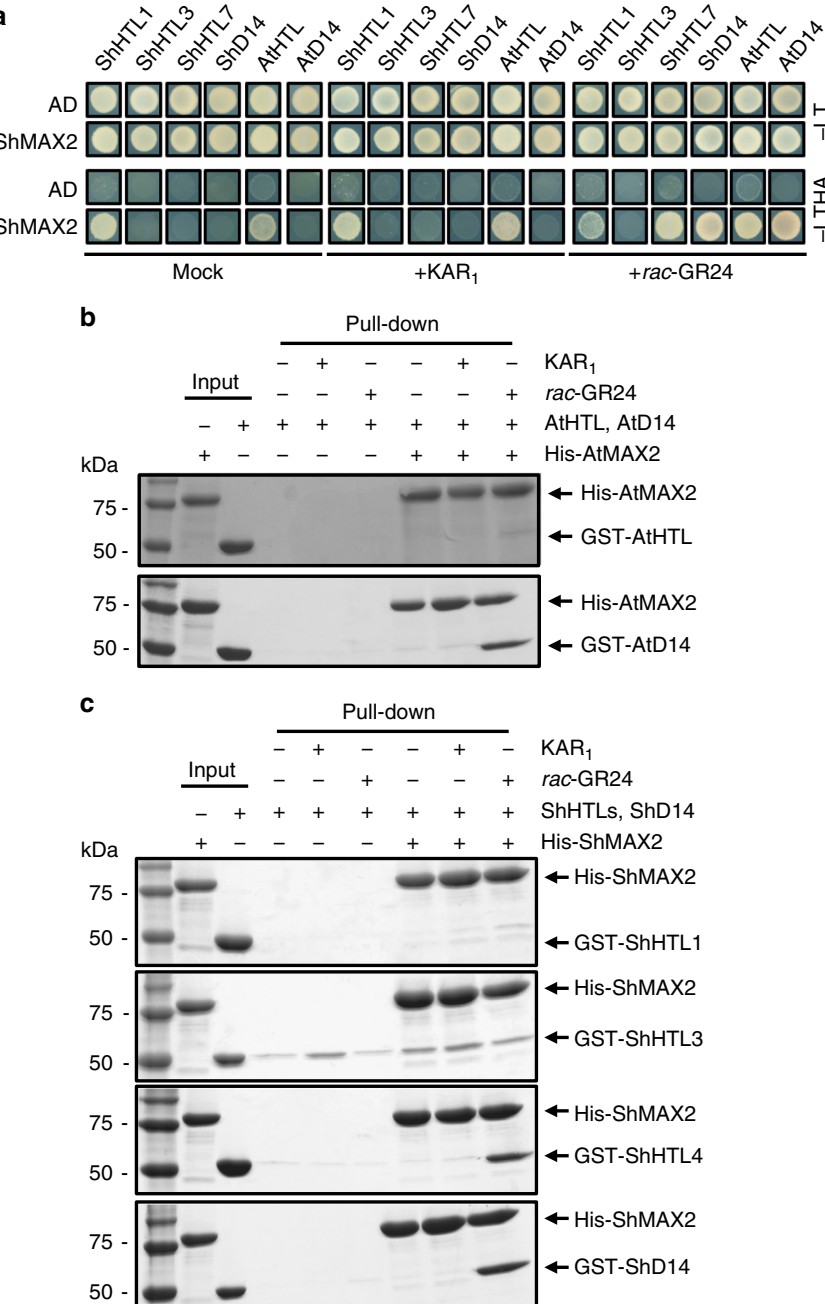

**Fig. 5** Interactions between HTLs/D14s and MAX2. **a** Interactions between HTL/D14 and ShMAX2, as determined by Y2H assays. Yeast cells were co-transformed with binding domain (BD)-fused ShHTL1, ShHTL3, ShHTL7, ShD14, AtHTL, or AtD14 and activation domain (AD)-fused ShMAX2. Interactions between bait and prey were detected in the absence or presence of 5 μM KAR$_1$ or 5 μM *rac*-GR24 in the control medium (SD/−Leu/−Trp; −LT) and the selective medium (SD/−Leu/−Trp/−His/−Ade; −LTHA). **b** Interactions between AtHTL or AtD14 and AtMAX2 according to the in vitro pull-down assays (detected by Coomassie Brilliant Blue (CBB) staining). His$_6$-tag-fused AtMAX2 was used as bait and AtHTL and AtD14 with GST-tags were used as prey. **c** Interactions between HTL/D14 and ShMAX2, as determined by the in vitro pull-down assays (detected by CBB staining). In vitro pull-down assays were performed using recombinant His$_6$-tag-fused ShMAX2 as bait. ShHTL1, ShHTL3, ShHTL4, and ShD14 with GST-tags were used as prey. Interactions were tested with or without 50 μM KAR$_1$ or 50 μM *rac*-GR24. The full gels are shown in Supplementary Fig. 15

observed for the ShHTL1–ShMAX2 and ShHTL1-AtMAX2 interactions. By contrast, for both AtHTL–AtMAX2 and AtHTL–ShMAX2 cotransformants, better growth was observed in the presence of *rac*-GR24 than the control and KAR$_1$. These results were consistent with those of the western blotting analysis in the present study and the results of a previous study showing a *rac*-GR24-induced AtHTL–AtMAX2 interaction in Y2H assays[27]. Therefore, it is suggested that ShHTL1 interacts with ShMAX2 or

AtMAX2 nonspecifically, while the interactions between AtHTL and ShMAX2 or AtMAX2 were induced in the presence of *rac*-GR24. Since AtHTL is able to hydrolyze GR24$^{ent−5DS}$, becomes thermally unstable in the presence of GR24$^{ent−5DS}$, similar to D14 upon binding to SL[26,34,37], and mediates a GR24$^{ent−5DS}$ response to regulate plant growth such as inhibition of hypocotyl elongation similar to SL response[35]; it is likely that AtHTL mediates the GR24$^{ent−5DS}$ response in a similar manner to the SL

response mediated by D14. Therefore, the results of our pull-down assays support the hypothesis that to some extent, AtHTL mediates the $GR24^{ent-5DS}$ response via the interaction with AtMAX2. Nonetheless, all the examined $KAR_1$-binding proteins failed to directly interact with the MAX2 proteins in a $KAR_1$-dependent manner.

## Discussion

Here, we present the crystal structures of the highly sensitive SL receptor, ShHTL7, as well as ShHTL1, ShHTL4, and ShD14 in *Striga*. A structural comparison revealed that the arrangement of helix αD1 and residues in the ligand-binding pocket are two major structural components that determine the pocket size, which in turn determine the KAR or GR24 specificity. The results of this study improve our understanding of the evolution of SL receptors in *Striga* and provide structural information for the further searches for SL analogs to combat *Striga*.

The evolutionary model of the different ligand recognition specificities of ShHTLs was improved by the structural findings of the present study (Fig. 6). ShHTL1 of the conserved clade is thought to be the ancestral ShHTL paralog[38]. ShHTL3 of the intermediate clade evolved from the ancestral HTL paralogs and developed the smallest binding pocket due to the presence of P136 and F190, which are proposed as the major structural bases for the ligand specificity of ShHTL3[31]. It is suggested that the divergent clade ShHTLs lost the ability to bind to KAR due to bulky residue 219, and they developed a highly sensitive subgroup (ShHTL4, ShHTL5, and ShHTL7) for SL binding likely via the evolution of F150 and other pocket-composing residues (e.g., residue 194). The divergent clade and intermediate clade are likely to be paralogs that evolved different biological functions during the duplication of the conserved clade.

Although it is well-known that MAX2 is required for both KAR signaling and SL signaling, with the latter requiring the D14–MAX2 interaction to transduce signals, there are few reports investigating the interaction between HTL and MAX2. In the present study, the interactions between ShHTLs and ShMAX2 were comprehensively determined. ShHTL4–ShMAX2, ShHTL7–ShMAX2, and ShD14–ShMAX2 interactions were induced in a *rac*-GR24-dependent manner. Weak AtHTL–AtMAX2 interactions were detected in the presence of *rac*-GR24. In contrast, AtHTL unexpectedly did not interact with AtMAX2 in a $KAR_1$-dependent manner. Consistent with this observation, other $KAR_1$-binding HTLs in *Striga*, ShHTL1, and ShHTL3, showed no direct interaction with ShMAX2 in the

presence of $KAR_1$, and the interaction between AtHTL and AtMAX2 has been questioned[27]. It is possible that MAX2 is involved in KAR signaling without any interaction with HTL or that a third component is required to enhance the HTL–MAX2 interaction, indicating a different pathway from SL signaling (Supplementary Fig. 14). Since DWARF53 (D53; a repressor of SL signaling) is able to interact with MAX2 and D14[39,40], members of the D53 family or its homologous SUPPRESSOR OF MAX2 1 LIKE (SMXL) family[41–44] (SMAX1; SMXL6/7/8) might be involved in enhancing the HTL–MAX2 interaction by forming a ternary complex. Alternatively, $KAR_1$ might fail to induce conformational changes in HTL similar to those observed in D14 in the presence of *rac*-GR24 and MAX2[17]. This speculation is supported by the evidence that the residues involved in the interaction with MAX2 are conserved between GR24-binding HTLs and the $KAR_1$-binding HTLs (Supplementary Fig. 2). In this case, $KAR_1$ might need to be converted into an active form in vivo for signal transduction to occur (Supplementary Fig. 14).

Our observation of $KAR_1$ binding to ShHTL1 is unexpected because previous studies have shown that ShHTL1 is non-responsive to exogenous $KAR_1$ in terms of *Arabidopsis* germination, and that $KAR_1$ fails to induce germination of *Striga*[12,13]. Although the $K_D$ value for KAR binding of ShHTL1 (77 μM) was higher than the physiological concentration (1 μM), this result might be because the fraction of active protein in the sample was lower than 100%, resulting in lower estimated affinity than real affinity. Moreover, similar results have been observed for some abscisic acid (ABA) receptors. For instance, PYRABACTIN RESISTANCE 1 (PYR1) has a $K_D$ for ABA of 97 μM[45]. Surprisingly, when PYR1 forms a complex with the co-receptor TYPE 2C PROTEIN PHOSPHATASE (PP2C), the binding affinities for ABA are in the nanomolar range. Thus, ShHTL1 might need other proteins or other factors to enhance the KAR binding affinity. It is also possible that ShHTL1 is involved in other signaling pathways instead of KAR-induced germination. KAR is a class of abiotic chemical derived from smoke, but *kai2-2* mutants and *max2* mutants in *Arabidopsis* both exhibit phenotypes of a reduced photomorphogenic response and increased primary dormancy, indicating that HTL and MAX2 are likely to be involved in the regulation of unknown endogenous phytohormones (termed KL or KAI2 ligand)[23,32,46]. Therefore, it is believed that AtHTL is involved in both KAR signaling and endogenous KL signaling[23,32,46]. Rather than the KAR receptor, ShHTL1 is likely to be the receptor for KL, an endogenous signaling molecule that shares structural similarity with KAR, in particular the butenolide ring. Indeed, conserved clade HTL orthologs exist in most Lamiids as a basal clade, implying their essential roles (e.g., KL pathway) in plants. Therefore, to fully understand the functions of HTL, further studies will be required to investigate the mysterious KL.

## Methods

**Overexpression and purification of recombinant proteins**. The coding regions of ShD14 (GenBank: KR013120.1), ShHTL1 (GenBank: KR013121.1), ShHTL4 (GenBank: KR013124.1), and ShHTL7 (GenBank: KR013127.1) were amplified by reverse transcription PCR, using the total RNA of *S. hermonthica* seeds. Recombinant ShHTL3 was produced as previously described[31]. ShD14 was cloned into the expression vector pET-49b (+) (Novagen). ShHTL1 and ShHTL4 were cloned into pGEX-6P-3 (GE Healthcare) with an N-terminal glutathione S-transferase (GST)-tag and a PreScission protease cleavage site. ShHTL7 was cloned into pET-28a (+) (Novagen) between the restriction enzyme sites *Nco*I and *Xho*I with a C-terminal His-tag. The recombinant proteins were expressed in the *E. coli* strain Rosetta (DE3) (Novagen). Cells were harvested by centrifugation at 5000 rpm for 10 min and stored at −80 °C until use. The harvested cells were resuspended in buffer A (20 mM Tris-HCl, pH 8.0, 300 mM NaCl and 1 mM DTT) for GST-tagged proteins or in buffer B (20 mM Tris-HCl, pH 8.0, 300 mM NaCl, 5% glycerol and 10 mM imidazole) for His-tagged ShHTL7, and they were lysed by sonication. The cell debris was removed by centrifugation at 40,000 × *g* for 30 min. The supernatant fractions of ShD14, ShHTL1, and ShHTL4 were then applied to Glutathione

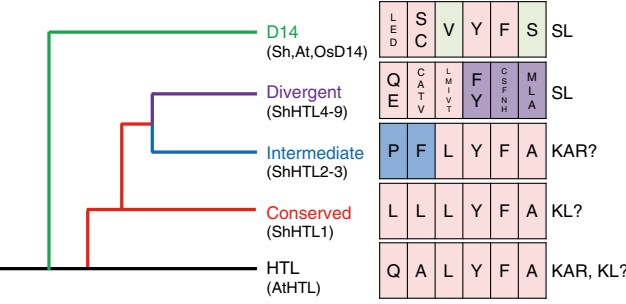

**Fig. 6** Model of the evolution of HTL/D14 proteins with different ligand specificities. The evolutionary model of ShHTLs was generated according to a previous model[13] with modifications (left). ShHTLs of the conserved, intermediate, and divergent clades are colored red, blue, and purple, respectively. The D14 clade is colored green. The residues determining their ligand specificities and the corresponding ligands are shown on the right. The residue number of ShHTL7 is shown on the top

Sepharose 4B resin (GE Healthcare). After washing with buffer A, the protease HRV3C was added to remove the GST-tag, and the untagged proteins were then eluted with buffer A. The eluate was further purified by loading onto a Resource Q or a Mono Q column (GE Healthcare), and elution was performed with a linear gradient of 0–1.0 M NaCl. ShHTL1 was further applied to a HiLoad 26/60 Superdex 75 pg column (GE Healthcare) to remove protein impurities. The supernatant fraction of ShHTL7 was applied to the Ni-NTA superflow resin (Qiagen), washed with buffer containing 20 mM imidazole, and eluted with buffer containing 200 mM imidazole. The eluate was then loaded onto a Mono Q column (GE Healthcare). All purified proteins were concentrated with Vivaspin 20 devices (10,000 MWCO PES) (Sartorius).

The DNA encoding ShMAX2 (GenBank: JX565467.1), AtMAX2 (GenBank: AF305597.1), and ASK1 (GenBank: AF059294.1) were cloned into the pFastBac Dual vector (Thermo Fisher Scientific) between the restriction enzyme sites *EcoRI* and *BamHI*. All the cloned genes were fused to an N-terminal His$_6$-tag following a FLAG-tag and PreScission protease cleavage site. *Spodoptera frugiperda* 9 (Sf9) insect cells were cultured in Sf-900 II SFM medium containing 50 units mL$^{-1}$ penicillin, 50 μg mL$^{-1}$ streptomycin, and 5% fetal bovine serum (FBS). For co-expression of MAX2 with ASK1, the baculovirus encoding ShMAX2 or AtMAX2 was co-infected in Sf9 insect cells with the baculovirus encoding ASK1. The cells were harvested after 3 days of incubation by centrifugation at 3000 rpm for 8 min. The harvested cells were resuspended in 20 mM Tris-HCl, pH 7.4, 300 mM NaCl, 10% glycerol, 1 mM DTT, and 10 mM imidazole and lysed by sonication. The cell debris was removed by centrifugation at 40,000 × *g* for 30 min. The protein was purified by Ni-NTA superflow resin, followed by a HiLoad 26/600 Superdex 200 pg column (GE Healthcare).

**Isothermal titration calorimetry**. The binding assays were performed using a MicroCal iTC$_{200}$ isothermal titration calorimeter (GE Healthcare). Prior to the ITC experiments, concentrated ShD14, ShHTL1, ShHTL4, and ShHTL7 were exchanged into a buffer consisting of 20 mM HEPES (pH 8.0) and 150 mM NaCl to remove dithiothreitol and were then adjusted to a final concentration of 150 μM. The sample cell was filled with protein solution (204 μL). Nineteen consecutive 2.0-μL aliquots of 3 mM KAR$_1$ (3-methyl-2*H*-furo [2,3-*c*] pyran-2-one) were injected into a prepared protein solution at 150 s intervals at 20 °C. The first injection volume was 0.4 μL, and the observed thermal peak was excluded from the data analysis. A negative control was obtained by titrating 3 mM KAR$_1$ into the buffer (20 mM HEPES, pH 8.0, 150 mM NaCl) in the same manner. Data fitting was performed in "one set of sites" mode, and $K_D$ values were calculated using Origin software provided by MicroCal.

**YLG hydrolysis assays**. The YLG assays were conducted as previously described[28] with minor modifications. Ten microliters of 0.1 mg mL$^{-1}$ ShHTL7 protein in a reaction buffer containing 20 mM HEPES, pH 7.0, 150 mM NaCl and 80 μL of reaction buffer were mixed in a 96-well black plate (Grenier). Ten microliters of 10 μM YLG was added to the solution and incubated at 25 °C for 1 h prior to measurement. The fluorescence intensity was measured via a SpectraMax® iD5 Multi-Mode Microplate Reader (Molecular Devices). Measurements were performed at room temperature under a 480 nm excitation wavelength, a 530 nm emission wavelength, a manual PMT of 500, and a read height of 1.00 mm. The fluorescence can be detected if YLG is hydrolyzed. The RFU (relative fluorescence unit) was calculated by subtracting the fluorescence of the buffer (without protein) and then dividing by the fluorescence of wild-type ShHTL7. The experiment was repeated 6 times independently.

**HPLC**. For the SL hydrolysis assays, 0.5 mL of PBS buffer (pH 7.3) containing 3.2 μM *rac*-GR24 (racemic mixture of a GR24$^{5DS}$ and GR24$^{ent-5DS}$) or 5DS was mixed separately with 1.6 μM purified ShHTLs and ShD14 and incubated for 30 min at 30 °C. Next, 50 μL of 1 M HCl was added to each reaction solution, and the reaction solutions were extracted 3 times with 300 μL of ethyl acetate. The ethyl acetate layers were combined and dried in vacuo and were then dissolved in 50 μL of methanol. For each layer, 5 μL was applied to the reverse-phase HPLC (Shimadzu LC-2030C). The analytical column was a CAPCELL CORE C18 (Φ 2.1 × 100 mm, Shiseido). The analytes were eluted under gradient conditions at a flow rate of 0.20 mL min$^{-1}$ with a linear ramp of methanol to 90% methanol at 19 min, which was maintained for 6 min before resetting to the original conditions. The amount of *rac*-GR24 or 5DS was calculated by the peak area (detection wavelength: 254 nm) at retention times ranging from 12.2–12.4 min with the regression equation obtained from the calibration curve produced using a dilution series of *rac*-GR24 or 5DS solutions. The percentage of hydrolyzed SL was calculated and used for statistical analyses. One-tailed unpaired Student's *t*- tests were used for comparison between buffer samples and enzyme samples.

**Yeast two-hybrid (Y2H) assay**. Full-length sequences of ShD14, ShHTL1, ShHTL3, ShHTL4, ShHTL7, and AtHTL were cloned into the pGBKT7 vector as bait (Clontech), and the coding regions of ShMAX2 and AtMAX2 were cloned into the pGADT7 vector as prey (Clontech). MAX2 was supposed to be unstable without ASK1. Therefore, AtASK1 was cloned into the N terminus of ShMAX2 and AtMAX2 with a 16-residue linker for co-expression. The bait and prey plasmids were co-transformed into the yeast (*Saccharomyces cerevisiae*) strain Y2HGold (Clontech) by the lithium acetate-mediated method. The presence of both plasmids was confirmed by growth on SD/-Leu/-Trp (-LT) plates. The interactions between bait and prey in the control medium (−LT) and selective medium (SD/−Leu/−Trp/−His/−Ade; −LTHA) in the absence or presence of 5 μM *rac*-GR24 (racemic mixtures) or 5 μM KAR$_1$ were detected. The plates were incubated for 5 days at 30 °C.

**In vitro pull-down assays**. Ni-NTA pull-down assays were performed with purified ShMAX2 or AtMAX2 as the bait and GST-ShHTLs, GST-ShD14, or GST-AtD14 as the prey. The molecular weight of the HTL/D14 proteins is nearly the same as that of ASK1; therefore, the HTL/D14 proteins were fused to GST, increasing their molecular weight for detection on SDS-PAGE gels. For this purpose, 25 μg of ShMAX2-ASK1 was mixed with 10 μg of GST-ShHTL1, GST-ShHTL3, GST-ShHTL4, or GST-ShD14 in the presence of 50 μM KAR$_1$, 50 μM *rac*-GR24, or a corresponding concentration of DMSO. After incubation at 4 °C for 30 min, the reaction mixtures were added to 100 μL of 50% (v/v) Ni-NTA superflow resin slurry that was pre-equilibrated with buffer containing 50 mM Bis–Tris, pH 6.8, 100 mM NaCl, 25 mM imidazole, and 10% glycerol. After further incubation at 4 °C for 30 min, unbound proteins were washed with buffer containing 50 mM Bis–Tris, pH 6.8, 100 mM NaCl, 50 mM imidazole, and 10% glycerol. The protein complexes were then eluted with 100 μL of elution buffer containing 50 mM Bis–Tris, pH 6.8, 500 mM NaCl, and 500 mM imidazole and subjected to SDS-PAGE analysis.

Samples of GST-ShHTL1 and GST-AtHTL were also subjected to western blotting after separation by SDS-PAGE. The proteins were transferred from the SDS-PAGE gel to a polyvinylidene fluoride (PVDF) membrane (Merck Millipore). The membrane was blocked by incubation in TBST buffer (50 mM Tris-HCl, pH 7.4, 138 mM NaCl, 2.7 mM KCl, and 0.1% (v/v) Tween-20) with the addition of 5% (w/v) bovine serum albumin for 1.5 h. Next, the membrane was incubated with anti-GST antibody as the primary antibody (71097–3; Novagen; diluted 1:10,000) at 25 °C for 1.5 h and then washed 3 times with TBST buffer. Anti-mouse antibody (32430; Thermo Scientific; diluted 1:4000) conjugated to horseradish peroxidase was used as the secondary antibody. After incubation with the secondary antibody at 25 °C for 1.5 h, the membrane was washed 3 times. For detection, SuperSignal West Femto Maximum Sensitivity Substrate (Thermo Scientific) was used as the chemiluminescence substrate and ImageQuant LAS 4000 mini (GE Healthcare) was used for imaging.

**Crystallization, data collection, and structure determination**. All crystallization trials were performed by the sitting-drop vapor diffusion method. Crystals of ShD14 were obtained by mixing 0.40 μL of purified protein solution (6.9 mg mL$^{-1}$) with 0.40 μL of reservoir solution containing 100 mM CAPS (pH 10.0), 200 mM NaCl, and 20% (w/v) PEG8000 and allowing the drop to equilibrate at 20 °C in 96-well Intelli-Plates (Art Robbins). Crystals of ShHTL1 were grown by mixing 0.45 μL of purified ShHTL1 protein solution (15.0 mg mL$^{-1}$) with 0.45 μL of reservoir solution containing 100 mM Tris (pH 8.5), 150 mM MgCl$_2$, and 23% (w/v) PEG6000 and allowing the drop to equilibrate at 5 °C. For crystallization of ShHTL4, 0.40 μL of protein solution (18.0 mg mL$^{-1}$) was mixed with 0.40 μL of reservoir solution containing 90 mM Tris (pH 9.0), 180 mM MgCl$_2$, 27% (w/v) PEG4000, and 1 mM GSH and incubated at 20 °C. Crystals of ShHTL7 were grown at 20 °C in 0.8 μL droplets obtained by mixing 0.50 μL of protein solution (7.5 mg mL$^{-1}$) and 0.30 μL of reservoir solution (90 mM Tris, pH 8.5, 180 mM MgCl$_2$, 27% (w/v) PEG5000 and 3.5% 1,4-dioxane).

A single crystal of ShD14 was picked up and soaked in the reservoir solution containing 20% (v/v) ethylene glycol as a cryoprotectant, and a diffraction data set was collected on beamline AR-NW12A at the Photon Factory (Tsukuba, Japan). The diffraction data for ShHTL1 were collected on beamline PF BL-5A at the Photon Factory (Tsukuba, Japan). The diffraction data for the ShHTL4 crystal were collected by using an in-house Rigaku R-AXIS VII imaging-plate detector (Rigaku, Japan). The diffraction data for the ShHTL7 crystal were collected on beamline PF BL-17A at the Photon Factory (Tsukuba, Japan). The diffraction data were indexed, integrated, and scaled using the XDS program[47]. The crystal structure of OsD14 (PDB code: 3VXK [https://doi.org/10.2210/pdb3VXK/pdb]) was used as a model for molecular replacement by MOLREP[48]. BUCCANEER[49] was applied for automatic model building. Iterative refinement cycles were performed using REFMAC5[50], PHENIX[51], and COOT[52]. Data collection and refinement statistics are presented in Table 1. The *B*-factors of the crystal structures were calculated by Baverage[53], and the volumes of the protein cavities were calculated using the CASTp server[54] with a probe radius of 1.4 Å. All structures were depicted by using PyMOL viewer (Version 1.8 Schrödinger, LLC). Sequence alignment was performed using CLUSTAL W[55] with default parameters, and the results were displayed via ESPript 3.0[56].

## Data availability
The atomic coordinates and structure factors of ShHTL1, ShHTL4, ShHTL7, and ShD14 have been deposited in the Protein Data Bank (PDB) under accession codes 5Z7W (https://doi.org/10.2210/pdb5Z7W/pdb), 5Z7X, (https://doi.org/10.2210/pdb5Z7X/pdb), 5Z7Y (https://doi.org/10.2210/pdb5Z7Y/pdb), and 5Z7Z (https://doi.org/10.2210/pdb5Z7Z/pdb),

respectively. All other data supporting the findings of this study are available within the manuscript and its supplementary files or are available from the corresponding author upon request.

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

## Acknowledgements

The synchrotron radiation experiments were performed on beamlines AR-NW12A, BL5A, and BL17A at the Photon Factory with the approval of the High Energy Accelerator Research Organization (Proposal No. 2015R-25). This work was supported by the

Platform for Drug Discovery, Informatics, and Structural Life Science from the Ministry of Education, Culture, Sports, Science, and Technology of Japan (MEXT) (M.T.), the Core Research for Evolutional Science and Technology (CREST) Program of the Japan Science and Technology Agency (JST) (T.A.), and a Grant-in-Aid for Scientific Research in Innovative Areas (JP17H05835 (T.M.) and JP18H04608 (H.N.)) from the Japan Society for the Promotion of Science (JSPS) and JSPS KAKENHI (JP15H05621 (T.M.) and JP26520303 (H.N.)). We thank Ikuo Takahashi for his help with the activity measurements.

## Author contributions

M.T. designed the research. Y.X. performed the biochemical experiments and collected X-ray diffraction data with T.M., S.N., A.N., Y.L. and H.N. performed the HPLC experiments. Y.X., T.M., U.O., H.I. and T.S. performed protein expression using the baculovirus-insect cell expression system. Y.X., T.M., S.N. and A.N. analyzed the data. Y.X., T.M., T.A. and M.T. wrote the paper. M.T. edited the manuscript.

## Additional information

**Competing interests:** The authors declare no competing interests.

