## [Peer Review File · Nature Communications]

Reviewers' comments:

Reviewer #1 (Remarks to the Author):

In this manuscript, the authors present a structural comparison of three types of HTL proteins found in the parasitic plant *Striga hermonthica*. These proteins have previously been shown to have different ligand specificities, primarily through transgenic studies. This study seeks to explain the structural differences that account for those ligand specificities. In addition, the authors evaluate the interaction of HTL proteins with their putative partner MAX2. This study provides valuable structural data and interesting hypotheses, but stops short of testing their hypotheses. Therefore, my primary critique is that a number of the conclusions made need to be less assertive because they require additional evidence. In addition, the authors need to acknowledge some of the existing literature that previously explored HTL-MAX2 interactions. I have proposed one or two minor experiments that would strengthen the results, but I consider my other comments, although many, to be easily remedied.

(line 26) "However, the recognition of strigolactone in the parasitic weed *Striga* is mediated by HTL instead of D14." This specifically refers to germination. The statement should be changed as D14 presumably perceives SL as well, which is supported by the evidence in this manuscript.

(line 61) The KAR-insensitivity of max2 mutants and kai2 mutants is strong evidence that both are involved in KAR responses. The established SL-induced D14-MAX2 interactions and the homology between D14 and HTL (including conservation of the residues involved directly in D14 interaction with MAX2) make a reasonable case for KAI2-MAX2 interactions. I agree that there is not a lot of direct evidence for KAI2-MAX2 interactions, but it isn't as speculative as the authors make it sound here. This paragraph should specify that htl mutants are KAR-insensitive, and the work of others showing HTL-MAX2 interactions (e.g. Y2H by Toh et al., 2014, *Chemistry & Biology*; rac-GR24-induced pulldowns between ShHTL7 and ShMAX2 by Yao et al., 2017, *Cell Research*) should be introduced and cited.

(line 64) I presume the authors intend to refer to perception of exogenous SL in parasite seeds rather than roots here.

(lines 66-69) The authors use the conserved, intermediate, and divergent clade classifications of Conn et al., 2015 but do not explain what this means (e.g. in terms of evolutionary rates, or more specifically, strength of selection) or where it originated. It's fine if they wish to use only the HTL nomenclature that was subsequently published for these genes, but they should point out the KAI2c, KAI2i, KAI2d synonyms at least once so that readers can convert between gene names in the different papers. Related to this, in (line 94) the authors should refer here to ShHTL3 also as ShKAI2iB, which they used in their prior publication that characterized this protein. Again, this reduces confusion in the literature.

(line 74) I would refer to YLG as a profluorogenic compound, as it does not become fluorescent until it is hydrolyzed.

(line 76 and elsewhere) Please use a different word than "responsivities."

(line 80) Specify/cite the previous in planta study of ShD14 referred to here.

(line 86) The authors should not claim priority here. Yao et al., 2017, for example, showed GR24-induced ShHTL7 interactions with ShMAX2 and SMAX1.

(lines 91-93, 106-107) Conn et al., 2016, *Frontiers in Plant Science* present evidence of ShHTL expression in transgenic *Arabidopsis* seedlings. It varies, but in the cases of ShHTL1 and ShHTL3 the expression can be comparable in some lines to wildtype AtKAI2. Regardless of expression level, the fact that the transgenic lines can show either rescue of *Arabidopsis kai2* mutant phenotypes (e.g. ShHTL1 in *kai2* seedlings and rosettes, although not seed germination), or positive responses to KAR or GR24 treatments demonstrates that they can function at least partially in *Arabidopsis*. This is not necessarily the case for ShHTL10 and ShHTL11, which do not show any effects in *Arabidopsis*.

Figure 1b might better be expressed as % hydrolyzed rac-GR24, so that the higher values on the graph correspond with higher catalytic activity.

(line 131) Citation for ShHTL3 should be for the authors' prior work (28), not 26.

(line 144) The concept of a highly sensitive HTL subclade is somewhat misleading. The authors' phylogeny indicates that a ShHTL4/5/7 subclade also contains ShHTL8 and ShHTL9, and potentially ShHTL6, depending on where one sets the subclade division. ShHTL6/8/9 all have Y150. I'd like to see a more in-depth phylogeny first, but it seems more likely that there was a Y150F substitution that independently evolved in 4/5 and 7. Potentially this results in higher SL sensitivity, but without any test of this hypothesis the authors need to be more cautious with their statement in lines 145-147.

(line 156) Specify which values.

(lines 157-158) Correlation does not equal causation. The evidence that this helix influences pocket size is good, but tests are needed to support the conclusion of altered ligand-binding specificity, particularly for the more subtle SL specificities that are shown by biochemical tests of the divergent HTL (Tsuchiya et al., 2015). It is worth highlighting that the smaller pocket size of ShHTL3 compared to ShHTL1 was not predicted by the modeling in Conn et al., 2015, which reinforces the value of this structural data.

(lines 166-169) It is notable that ShHTL7 was shown to have similar *in vitro* affinity for SLs as other divergent HTLs (Tsuchiya et al., 2015), which was quite different from the responses seen in transgenic lines (Toh et al., 2015). I would be more cautious about how the ShHTL7 pocket size and residues influence GR24 sensitivity when it could be due to alternative explanations, such as compatibility with *Arabidopsis* proteins or expression levels, which the authors previously invoked to discuss the (in)efficacy of other HTLs.

(lines 170..) Indicate the rationale for these particular amino acid substitutions. Was it to mimic ShHTL3? Provide the K_d values based on the ITC data for the two variant proteins. Also, effects on YLG activity are not particularly relevant (this is demonstrated by the authors' experiments here on ShD14) for indicating SL recognition, as they are different molecules. For this same reason, (line 174) is incorrect to equate YLG hydrolysis with SL binding/hydrolyzing ability. However, if the authors would examine the IC_{50} of GR24 on YLG hydrolysis, similar to Tsuchiya et al., 2015, it may provide useful information about the effects of these mutations. Ideally, an *in vivo* test would be used to determine changes in ligand sensitivity, but this would require substantial effort and time that is beyond a reasonable revision.

(line 179) Given the data in this manuscript, "KAR-binding" HTLs seems more appropriate than "KAR-recognizing"

(line 182-183) AtHTL has enough space to bind and hydrolyze GR24-ent5DS. It seems that fit is more the issue than pocket size for accommodating GR24, at least for AtHTL. Which data on SL binding for

these proteins is being referred to? The hydrolysis rate of rac-GR24?

(line 185) Indicate the pocket volume of AtHTL for comparison. Is it a dramatic difference?

(line 196) What is the evidence that V142 is highly conserved? Which collection of protein sequences have been surveyed to determine this?

(line 198) GR24-5DS has a natural 2'R stereochemical configuration, but is not natural itself. This could be clearer.

(line 199-200) Refer to the evidence that GR24-ent5DS is likely to avoid the steric clash.

Please specify how the amino acid numbers are decided. There is some variation in the length of D14 and HTL proteins, so it would be useful to know if the numbers refer to residues in a specific protein sequence or a D14 or HTL consensus, etc. This should make it easier to compare to residues highlighted in prior studies.

(line 217-222) There is an argument that can be made that D14 in parasites likely functions similarly to D14 in other angiosperms (strong sequence conservation, maintenance of a single copy, parsimony, inability of AtD14 to rescue *kai2* germination), but I don't think that ShD14-ShMAX2 interaction or unchanged expression of ShD14 during germination are evidence that it doesn't function in germination. I would rephrase the logic here.

(Figure 5) CBB stain isn't defined or noted in the methods, although I'm guessing it's Coomassie Brilliant Blue staining. Please clarify.

(line 242-243) As the authors just pointed out in (lines 197-199), AtHTL responds to a GR24 molecule with an unnatural stereochemical configuration. Therefore, AtHTL is not mediating a SL response, although it does mediate a GR24 response.

(Figure 5) The Western blot data for AtHTL-AtMAX2 in Supplementary Figure 8 support a rac-GR24-induced interaction in pulldown. In light of this, it looks like the Y2H data in Figure 5a might show better growth for AtHTL-AtMAX2 in the presence of rac-GR24 than the control. It would be worth trying this particular assay again with a dilution series and replicate colonies to see if this is real and consistent with the Western results. LacZ staining may also be useful as a complementary reporter to growth, similar to Toh et al., 2014, *Chemistry and Biology*.

(line 249-253, 260-264) The structural basis for high sensitivity of ShHTL7 has not been revealed, although the authors have certainly come up with some interesting hypotheses for that basis. Likewise, the basis for KAR or SL specificity or ShHTL3 specificity have not been determined, although some factors that may influence that have been identified. The significance of F150 for sensitivity has not been directly tested either. Essentially, these conclusions need to be softened as they are still untested hypotheses.

Reviewer #2 (Remarks to the Author):

In this manuscript, Xu et al. analyze the D14/HTL family of SL- and KAR-binding proteins in the parasitic plant *Striga hermonthica* (Sh). *Striga* causes devastating crop losses in sub-Saharan Africa. They sense host plants by their release of SL, which function among others as chemoattractants for mycorrhiza-forming symbionts. In non-parasitic plants SL are sensed by D14 receptors, and

germination-stimulating KAR by D14-homologous HTL proteins. In contrast, in *Striga*, SL are sensed by D14 and a divergent clade of HTL proteins, and stimulate germination, while KAR are bound by a smaller subclade of HTL proteins more closely related to ATHTL binds, but fail to induce germination. By employing a combination of compound- and protein-protein interaction assays with ShHTL/ShD14 structural analysis, the authors report the following major findings:

1. ShHTL1, which has been reported to be unresponsive to KAR, binds KAR with similar affinity as AtHTL and ShHTL3. Surprisingly, although all three proteins bind KAR, and the signal transducer MAX2 is required for KAR-induced germination in non-parasitic plants, KAR does not mediate HTL-MAX2 interaction. Therefore KAR-signaling shows a major difference to SL signaling, as SL function by inducing D14-MAX2 interaction.
2. The authors determined crystal the structures of ShHTL1, ShHTL4, ShHTL7, and ShD14. Structural comparison of D14 and HTL proteins from parasitic and non-parasitic plants identified pocket size, determined by the bulkiness of pocket-lining residues, as selectivity and affinity determinants. This allowed the authors to propose a "code" of a small number of aa positions that define ligand interactions and that can be utilized for the development of *Striga*-combatting "suicide germination stimulants".

Overall, the experiments are of high quality and convincing, and of interest to a broad group of researchers studying SL/KAR signaling.

Minor points: Please correct the following overreaching statements. Lines 103-105: The result that ShHTL1 binds KAR in vitro is NOT inconsistent with the reported lack of KAR responsiveness in cross-complementation assays. Lines 157: please replace "indicate" with "suggests". Line 174: The authors have NOT provided evidence that their mutants reduced SL binding. Instead they have shown that the mutants reduce SL hydrolysis, which is suggestive of reduced binding, but by no means proves it. In fact, the authors should determine binding affinities by ITC. Also, GR245DS is not a "natural" SL (it is a synthetic one). Rather, it is biologically active in inducing SL responses, whereas GR24ent-5DS is not.

Reviewer #3 (Remarks to the Author):

The discovery that genomes of parasitic plants of the Orobanchaceae encode a family of HTL/KAI2 proteins that can function in strigolactone (SL) and karrikin (KAR) signalling when transferred into *Arabidopsis* has stimulated much research on the structure and function of these proteins. Their role in the perception of host plant strigolactones by seeds of parasitic plants is a very important area of plant biology.

The HTL genes fall into three clades: Conserved (HTL1), Intermediate (HTL2 and 3) and Divergent (HTL4-11). There is also one putative D14 gene reported in *Striga hermonthica* which is expected to encode the putative strigolactone receptor involved in host plant development.

This new manuscript of Xu et al. evaluates KAR and SL binding activity of three *Striga* HTLs (ShHTLs) and also ShD14.

These authors have previously published data on HTL3 from the Intermediate group and provided evidence that it recognises karrikin, consistent with the karrikin response of transgenic *Arabidopsis* expressing HTL3.

The authors now show that HTL1 (Conserved clade) could bind karrikin in Isothermal Calorimetry (ITC) assays even though previous reports indicate that it did not apparently respond to karrikin in

transgenic Arabidopsis. Xu et al now also show that HTL1 from Arabidopsis binds to karrikin in ITC assays.

Xu et al show that HTL4 and 7 (Divergent clade) did not bind karrikin. This is consistent with lack of karrikin response in transgenic Arabidopsis (previous reports).

They further showed that ShHTL1 does not hydrolyse the synthetic strigolactone GR24, whereas HTL4, 7 and 14 do. These results are consistent with data from previously published research indicating that HTLs in the Intermediate clade probably recognise strigolactones.

The authors determined crystal structures of HTL1, 4 and 7, plus ShD14 in an attempt to explain responses of these proteins to different ligands.

They further showed that HTL1 does not bind F-box protein ShMAX2. Similar results are reported for ShHTL3.

In contrast D14, HTL4 and HTL7 hydrolyse rac-GR24 slowly, and bind to ShMAX2. These results are largely consistent with other published results.

The research makes a useful contribution to the field of study but does not provide any major new insight.

MAIN POINTS

1. A limitation of this research is that the endogenous or natural ligands for ShHTL proteins are not known. This is equally true of the D14 clade which respond to endogenous strigolactones, since the active ligands have not been identified, even in Arabidopsis.

While the use of a racemic mixture of GR24 isomers is convenient, and is commonly employed, it could be misleading, when aiming to elucidate the ligand binding specificity of HTL proteins. It would be far better to use stereospecific natural strigolactones such as strigol or orobanchol instead of synthetic analogues.

2. There is increasing evidence that karrikins are not active ligands, but may be converted into active compounds in planta. Xu et al., acknowledge this. For example two studies of crystal structures of AtHTL/AtKAI2 protein with KAR1 present data showing KAR1 in different positions in the active site pocket, neither of which is closely associated with the active site amino acid residues comprising the catalytic triad. Furthermore, studies using Differential Scanning Fluorimetry (DSF) can readily detect interaction of (+)-GR24 with AtD14 and interaction of (-)-GR24 with AtHTL/AtKAI2, but no interaction with KAR (ref 35). Such results cast doubt on the validity of using KAR to study interactions with HTL proteins.

In order to observe interaction the ITC analysis presented here is performed with 3mM KAR1 at a 20-fold molar excess of KAR1 relative to protein. Values for K_d are 77 μ M for ShHTL1, 78 μ M for ShHTL3 (previous publication), and 129 μ M for AtHTL1. These values are very high. KAR1 is typically active at 1 μ M in Arabidopsis, or at much lower concentrations with some karrikin-responsive species. For comparison, a K_d value of 0.3 μ M is reported for rac-GR24 binding to AtD14 (ref 15).

The authors acknowledge (P5 line 16-21) that ShHTL1 expressed in Arabidopsis does not respond to karrikins. They suggest that it is possible that the heterologous protein is not expressed adequately, or another explanation is that the heterologous protein does not interact functionally with protein partners required for signal transduction. While possible, there are two arguments against these

explanations; Firstly, seeds of *S. hermonthica* do not respond to KAR1. Secondly a KAI2/HTL-type protein from *Selaginella* can complement the *Arabidopsis kai2-1* mutant but does not respond to karrikins, GR24, debranones or carlactone (ref 35).

3. Throughout the manuscript the authors should specify that they use rac-GR24 (not simply GR24). The different stereoisomers of GR24 have different biological activities in *Arabidopsis* and behave differently in hydrolysis and binding assays.

4. The authors should not assume that GR24 and natural strigolactones behave the same, and so should be extremely careful not to equate GR24 with 'SL'. For example P6 lines 11-12.

MINOR POINTS

Abstract. 2-3. There is little evidence that SL is a germination stimulant in *Arabidopsis* and none in rice.

Intro p3 line 14. Use of the word 'endogenous' is misleading because the endogenous SL is unknown and there is no endogenous karrikin.

16-18. Although it is correct to say that D14 requires MAX2 as a signalling component, this is not so for HTL/KAI2 because there is little evidence for interaction between HTL/KAI2 and MAX2 (as this manuscript confirms). Lines 20-24 are accurate. Lines 16-18 should be modified.

18-20. The attack on SL by D14 is not truly hydrolysis because the final step is not completed.

P4 line 2. 'Perception in roots' is confusing. Perception by seeds?

Line 8 (and elsewhere throughout the manuscript). Genes are not transformed into plants; Plants are transformed with genes.

9-10. The identity of SLs should be given since results can vary depending on which compounds are used.

12. Define YLG

15-23. This paragraph is a summary of results, but should explain the aims. Lines 17-18 and 21-23 are cryptic, not informative.

P5. Line 3-4. Re-word and explain fully. Mutants are organisms resulting from mutations.

P6 line 8 'hydrolysis assays'

Line 9. 'Data not shown'. Data should be shown.

Line 10-11. 'It has been suggested that ShD14 also serves as an SL receptor.' Provide reference.

Line 12. 'hydrolytic activity towards SL. ShHTL4, ShHTL7 and ShD14 were able to hydrolyze SL' Authors should be specific and refer to rac-GR24 instead of SL, since no natural SLs have been investigated.

P8. Lines 10. It is confusing to refer to double and triple mutants because only a single gene is mutated. Better to refer to mutations which produce HTL protein with two or three amino acid substitutions.

P11. Lines 7-9. 'Since AtHTL is able to hydrolyze GR24 (30,31,35) and is involved in GR24-inducing germination (13), the results of our pull-down assays support the hypothesis that AtHTL mediates SL response, to some extent, via the interaction with AtMAX2.' I think this interpretation is highly questionable because the published genetic evidence is that SL does not induce signalling via AtHTL. One stereoisomer of GR24 is active. While Atmax2 mutants do not respond to KAR, this does not provide evidence for functional interaction between MAX2 and HTL.

P12, line 9. We need to be more cautious about MAX2 being 'involved' in KAR response since max2 mutants might simply be pleiotropic.

Overall the Discussion should be more cautious about the use of unnatural chemicals (KAR and rac-GR24) to study ligand specificity of the HTL and D14 proteins.

We appreciate the reviewers' careful reading and valuable comments and suggestions. The following are point-by-point responses to the reviewers' comments on our manuscript entitled "Comprehensive structural studies reveal the evolution of strigolactone/karrikin selectivity of HTL and D14 in *Striga*". We accepted most of the suggestions and improved the manuscript accordingly. We have attached the revised manuscript in which the revised portions of the text are marked in red.

Reviewers' comments:

Reviewer #1 (Remarks to the Author):

[Comment 1]

In this manuscript, the authors present a structural comparison of three types of HTL proteins found in the parasitic plant *Striga hermonthica*. These proteins have previously been shown to have different ligand specificities, primarily through transgenic studies. This study seeks to explain the structural differences that account for those ligand specificities. In addition, the authors evaluate the interaction of HTL proteins with their putative partner MAX2. This study provides valuable structural data and interesting hypotheses, but stops short of testing their hypotheses. Therefore, my primary critique is that a number of the conclusions made need to be less assertive because they require additional evidence. In addition, the authors need to acknowledge some of the existing literature that previously explored HTL-MAX2 interactions. I have proposed one or two minor experiments that would strengthen the results, but I consider my other comments, although many, to be easily remedied.

[Response]

Thank you very much for your valuable comments and constructive suggestions. We have performed experiments that you suggested and revised our manuscripts as follows.

[Comment 2]

(line 26) "However, the recognition of strigolactone in the parasitic weed *Striga* is mediated by HTL instead of D14." This specifically refers to germination. The statement should be changed as D14 presumably perceives SL as well, which is supported by the evidence in this manuscript.

[Response]

We agree with the reviewer. Our former expression "In non-parasitic plants such as Arabidopsis and rice, two plant germination stimulants, strigolactone and karrikin..." is confusing. Therefore, these sentences in lines 22-24 have been rewritten as follows:

"HYPOSENSITIVE TO LIGHT (HTL) mediates the perception of both karrikin and the germination stimulant of the parasitic weed *Striga*, strigolactone. Strigolactone is also a phytohormone that needs DWARF14 (D14) as a receptor."

[Comment 3]

(line 61) The KAR-insensitivity of max2 mutants and kai2 mutants is strong evidence that both are involved in KAR responses. The established SL-induced D14-MAX2 interactions and the homology between D14 and HTL (including conservation of the residues involved directly in D14 interaction with MAX2) make a reasonable case for KAI2-MAX2 interactions. I agree that there is not a lot of direct evidence for KAI2-MAX2 interactions, but it isn't as speculative as the authors make it sound here. This paragraph should specify that htl mutants are KAR-insensitive, and the work of others showing HTL-MAX2 interactions (e.g. Y2H by Toh et al., 2014, Chemistry & Biology; rac-GR24-induced pulldowns between ShHTL7 and ShMAX2 by Yao et al., 2017, Cell Research) should be introduced and cited.

[Response]

We have added the description that *htl* mutants are KAR-insensitive and the information of previous reports about HTL-MAX2 interactions (lines 59-60). We have also deleted the following sentence: "However, there is no direct evidence of whether MAX2 is involved in KAR signaling by interacting with HTL." *Rac*-GR24-induced interaction between ShHTL7 and ShMAX2 has been stated (lines 246-247) and the previous work by Yao *et al.* has been cited (ref. 29).

[Comment 4]

(line 64) I presume the authors intend to refer to perception of exogenous SL in parasite seeds rather than roots here.

[Response]

We intend to refer to perception of exogenous SL in parasite seeds and we have revised the corresponding words (line 62).

[Comment 5]

(lines 66-69) The authors use the conserved, intermediate, and divergent clade classifications of Conn et al., 2015 but do not explain what this means (e.g. in terms of evolutionary rates, or more specifically, strength of selection) or where it originated. It's fine if they wish to use only the HTL nomenclature that was subsequently published for these genes, but they should point out the KAI2c, KAI2i, KAI2d synonyms at least once so that readers can convert between gene names in the different papers. Related to this, in (line 94) the authors should refer here to ShHTL3 also as ShKAI2iB, which they used in their prior publication that characterized this protein. Again, this reduces confusion in the literature.

[Response]

Your valuable comments help improve the readability of the manuscript. We have added explanation of classifications of conserved, intermediate, and divergent clades (lines 65-69). The relationship of synonyms of KAI2c, KAI2i, KAI2d and ShHTL1-11 has been annotated. We have also included the name "ShKAI2iB" (lines 107-108), which was used in our previous report.

[Comment 6]

(line 74) I would refer to YLG as a profluorogenic compound, as it does not become fluorescent until it is hydrolyzed.

[Response]

We have replaced the word "fluorogenic" with "profluorogenic" (line 77).

[Comment 7]

(line 76 and elsewhere) Please use a different word than "responsivities."

[Response]

We have replaced the word "responsivities" with "responsiveness" (lines 80 and 103).

[Comment 8]

(line 80) Specify/cite the previous in planta study of ShD14 referred to here.

[Response]

We used ambiguous expression in the former version of manuscript. In fact, there is no report of *in planta* study of ShD14 although YLG hydrolysis study of ShD14 has been reported. Therefore, we have rewritten the related sentences as follows (lines 82-86):

"Our results showed that ShHTL1 was able to bind KAR, despite no KAR-dependent germination responses in a previous *in planta* study¹³. We also showed that ShD14 was capable of hydrolyzing SL, which was unexpected because no hydrolytic activity of ShD14 towards YLG has been detected in a previous study²⁷."

[Comment 9]

(line 86) The authors should not claim priority here. Yao et al., 2017, for example, showed GR24-induced ShHTL7 interactions with ShMAX2 and SMAX1.

[Response]

We have revised the related sentences as follows (lines 94-99):

"Furthermore, only ShHTL7 has been examined about its interaction with the putative downstream signaling component ShMAX2 (*S. hermonthica* MAX2)²⁹. However, it remains unclear whether ShHTLs of different clades are able to interact with ShMAX2 to transduce signals like D14-MAX2 in *Arabidopsis*. Therefore, interactions between other members of ShHTLs and ShMAX2 were determined comprehensively in the present study."

[Comment 10]

(lines 91-93, 106-107) Conn et al., 2016, *Frontiers in Plant Science* present evidence of ShHTL expression in transgenic *Arabidopsis* seedlings. It varies, but in the cases of ShHTL1 and ShHTL3 the expression can be comparable in some lines to wildtype AtKAI2. Regardless of expression level, the fact that the transgenic lines can show either rescue of *Arabidopsis kai2* mutant phenotypes (e.g. ShHTL1 in *kai2* seedlings and rosettes, although not seed germination), or positive responses to KAR or GR24 treatments demonstrates that they can function at least partially in *Arabidopsis*. This is not necessarily the case for ShHTL10 and ShHTL11, which do not show any effects in *Arabidopsis*.

[Response]

As you pointed out, the expression of ShHTL1 (ShKAI2c) and ShHTL3 (ShKAI2i) in transgenic lines have been confirmed and ShHTL1 can rescue some phenotypes of *kai2* mutant but not germination phenotype. Therefore, the following sentences have been deleted.

"It is uncertain whether the *ShHTL* homologs were expressed in the transgenic plants" and "It is possible that ShHTL1 was not expressed in *Arabidopsis* in the previous report".

The statement about the transgenic lines of ShHTL1 rescuing some *Arabidopsis kai2* mutant phenotypes has been added (lines 117-120):

"Our results of ShHTL1 binding KAR₁ was unexpected because it has been reported that ShHTL1 was responsive to neither KAR₁ nor *rac*-GR24 in cross-species complementation assays in terms of seed germination but rescued some other phenotypes^{13,31} (discussed later)."

The statement of "it is uncertain whether the *ShHTL* homologs were expressed in the transgenic plants" has also been revised (lines 105 and 106) to "it is still unknown whether the *ShHTL* homologs bind certain ligands *in planta*".

[Comment 11]

Figure 1b might better be expressed as % hydrolyzed *rac*-GR24, so that the higher values on the graph correspond with higher catalytic activity.

[Response]

Figure 1b has been revised as suggested.

[Comment 12]

(line 131) Citation for ShHTL3 should be for the authors' prior work (28), not 26.

[Response]

We have corrected the reference number.

[Comment 13]

(line 144) The concept of a highly sensitive HTL subclade is somewhat misleading. The authors' phylogeny indicates that a ShHTL4/5/7 subclade also contains ShHTL8 and ShHTL9, and potentially ShHTL6, depending on where one sets the subclade division. ShHTL6/8/9 all have Y150. I'd like to see a more in-depth phylogeny first, but it seems more likely that there was a Y150F substitution that independently evolved in 4/5 and 7. Potentially this results in higher SL sensitivity, but without any test of this hypothesis the authors need to be more cautious with their

statement in lines 145-147.

[Response]

Although all of ShHTL4-9 are members of highly sensitive HTL subclade, ShHTL4/5/7 are extremely sensitive to natural SLs (strigol, 5-deoxystrigol, 2'-epi-5-deoxy strigol and sorgolactone) with respect to germination according to previous work by Toh *et al.* and ShHTL4/5/7 have a F150 instead of Y150 in ShHTL6/8/9. We referred to ShHTL4/5/7 as a subclade, which is a phylogenetic term and is inappropriate to be used in this case. What we were supposed to refer to is a subgroup. The related words have been corrected (lines 153, 311 and 696).

We have proposed a structural basis for the effect of F150 and Y150 on the ligand-binding site. However, the relation between higher SL sensitivity and 150th residue has not been tested, and thus we have softened our conclusion as follows (lines 158-159):

"Therefore, it is speculated that owing to the change in the 150th residue..."

[Comment 14]

(line 156) Specify which values.

[Response]

We have specified the values by referring to Fig. 4a and revised the related sentence as follows (lines 174-176):

"It is apparent the more the helices α D1 shift outward (Fig. 3b), the larger sizes the ligand-binding pockets become (Fig. 4a). For example, ShHTL7 with the largest pocket has the helix α D1 that tilts the greatest away from the entrance of the pocket."

[Comment 15]

(lines 157-158) Correlation does not equal causation. The evidence that this helix influences pocket size is good, but tests are needed to support the conclusion of altered ligand-binding specificity, particularly for the more subtle SL specificities that are shown by biochemical tests of the divergent HTL (Tsuchiya *et al.*, 2015). It is worth highlighting that the smaller pocket size of ShHTL3 compared to ShHTL1 was not predicted by the modeling in Conn *et al.*, 2015, which reinforces the value of this structural data.

[Response]

As you pointed out, our crystal structures suggest that helix α D1 constituted the entrance to the ligand-binding pocket and the orientation of helix α D1 influenced the pocket size. To examine the relation between the orientation of helix α D1 and ligand-binding specificity, we considered swapping helix α D1 from one ShHTL to another ShHTL as a direct experiment. However, it is difficult to control the orientation of the swapped helix α D1 because the orientation depends on other factors such as interaction with helices α D2, α D3 and α D4. On the other hand, we proved that changing the pocket size of ShHTL7 using point mutation (lines 208-215) could alter its ligand specificity, which supports the hypothesis that the change in pocket size can alter ligand specificity. However, our experiments do not provide the direct evidence on the effect of helix α D1. Therefore, the related sentence has been revised as follows (lines 176-179):

"Since helix α D1 constituted a part of the entrance to the ligand-binding pocket, it is suggested that orientation of helix α D1 affected the size of ligand-binding pockets, and thus helix α D1 might be an important structural determinant of their ligand-binding specificity."

As suggested, the smaller pocket size of ShHTL3 compared to ShHTL1 was not predicted by the modeling. In addition, our structural data revealed the distinct orientation of helix α D1. The related statement has been added (lines 167-171).

[Comment 16]

(lines 166-169) It is notable that ShHTL7 was shown to have similar in vitro affinity for SLs as other divergent HTLs (Tsuchiya et al., 2015), which was quite different from the responses seen in transgenic lines (Toh et al., 2015). I would be more cautious about how the ShHTL7 pocket size and residues influence GR24 sensitivity when it could be due to alternative explanations, such as compatibility with *Arabidopsis* proteins or expression levels, which the authors previously invoked to discuss the (in)efficacy of other HTLs.

[Response]

We agree with your comments. ShHTL7 is highly sensitive to YLG and other natural SLs according to the previous paper (Tsuchiya *et al.*, 2015). Although ShHTL7 is sensitive to picomolar concentrations of SL in *Arabidopsis*, compatibility with *Arabidopsis* proteins or expression levels should be taken into consideration when it comes to the responses in transgenic lines. Therefore, we have deleted the following sentence:

"Intriguingly, ShHTL7 has been reported to be the most sensitive SL receptor in response to GR24."

Moreover, we have discussed how the ShHTL7 pocket size and residues influence GR24 sensitivity based on the mutation of ShHTL7 by introducing larger residues in the ligand-binding pocket (lines 208-215).

[Comment 17]

(lines 170..) Indicate the rationale for these particular amino acid substitutions. Was it to mimic ShHTL3? Provide the K_d values based on the ITC data for the two variant proteins. Also, effects on YLG activity are not particularly relevant (this is demonstrated by the authors' experiments here on ShD14) for indicating SL recognition, as they are different molecules. For this same reason, (line 174) is incorrect to equate YLG hydrolysis with SL binding/hydrolyzing ability. However, if the authors would examine the IC_{50} of GR24 on YLG hydrolysis, similar to Tsuchiya et al., 2015, it may provide useful information about the effects of these mutations. Ideally, an in vivo test would be used to determine changes in ligand sensitivity, but this would require substantial effort and time that is beyond a reasonable revision.

[Response]

We generated mutation of L190F/C194F and L124F/T190F/C194F of ShHTL7 by introducing the residue with larger side chain so as to make the pocket smaller by partially mimicking ShHTL3. Based on the structural comparison of the SL/KAR-binding pocket, we focused on the bulkier residues specific to the KAR-binding HTLs. Especially, the size of 190th residue is considerably different between ShHTL1/ShHTL3 (Leu/Phe) and D14s/ShHTLs of the divergent clade (Ser/Cys/Thr). In addition, the 124th and 194th residues are aromatic residues (Phe/Tyr) for D14s, ShHTL1 and ShHTL3 but ShHTLs of the divergent clade adopt the smaller residues at the same position. To indicate the rationale for these particular amino acid substitutions, the sentences for the mutation experiments has been moved after the description about the structural comparison of the SL/KAR-binding pocket (lines 208-215).

It is inappropriate to use "SL binding/hydrolyzing ability" when only hydrolytic activity towards YLG has been examined. Therefore, we performed YLG competition assays using *rac*-GR24 and calculated the inhibition constant $K_i^{rac-GR24}$ (the reciprocal of the binding affinity of the inhibitor to the enzyme), which is more appropriate for representing the binding affinity of *rac*-GR24 to the enzyme when K_m^{YLG} values of the wild type and the mutants differ. According to our results, $K_i^{rac-GR24}$ of ShHTL7 is smaller than ShHTL7^{L190F/C194F} and ShHTL7^{L124F/L190F/C194F}, indicating that these mutations have reduced *rac*-GR24 binding affinity. We have revised Supplementary Fig. 7

and revised the manuscript as follows (lines 212-215):

"The inhibition constant $K_i^{rac-GR24}$ in YLG competition assays is used to indicate the *rac*-GR24 binding affinity of ShHTL7 and it is shown that T190F/C194F mutation led to 2-fold decrease and L124F/T190F/C194F mutation led to more than 100-fold decrease in *rac*-GR24 binding affinity (Supplementary Fig. 7)."

[Comment 18]

(line 179) Given the data in this manuscript, "KAR-binding" HTLs seems more appropriate than "KAR-recognizing"

[Response]

We have replaced "KAR-recognizing" with "KAR-binding". (line 194)

[Comment 19]

(line 182-183) AtHTL has enough space to bind and hydrolyze GR24-ent5DS. It seems that fit is more the issue than pocket size for accommodating GR24, at least for AtHTL. Which data on SL binding for these proteins is being referred to? The hydrolysis rate of *rac*-GR24?

[Response]

Biochemical (Flematti *et al.*, 2016) and genetic study (Scaffidi *et al.*, 2014) have shown that AtHTL is able to bind and hydrolyze GR24^{ent-5DS}. Therefore, we have corrected the related sentence to "ShHTL1 and ShHTL3 can bind KAR₁ but not *rac*-GR24 (Supplementary Fig. 6)." (lines 198) by deleting "AtHTL". In the case of AtHTL, both pocket size and fit seem to be important for ligand binding. Pocket size is crucial for their KAR/SL specificity. When the size is suitable, fit (without steric clash) becomes important as well.

Rac-GR24 binding and hydrolysis (but not SL) activity of ShHTL3 have been tested and no binding or hydrolysis has been detected in our previous study (ref. 30). We have detected no *rac*-GR24 hydrolysis ability of ShHTL1 as shown in Fig. 1b. We have also added ITC data in Supplementary Fig. 6, showing that ShHTL1 has no detectable binding affinity to *rac*-GR24.

[Comment 20]

(line 185) Indicate the pocket volume of AtHTL for comparison. Is it a dramatic difference?

[Response]

We have added the calculated pocket volumes of ShHTL1 and AtHTL for comparison (lines 199 and 200). The ligand-binding pocket size of AtHTL (411 Å³) is larger than ShHTL1 (321 Å³) and ShHTL3 (274 Å³) but smaller than D14 group (e.g. 510 Å³). AtHTL is just big enough to bind GR24, and thus the difference between ShHTL1 and AtHTL is dramatic.

[Comment 21]

(line 196) What is the evidence that V142 is highly conserved? Which collection of protein sequences have been surveyed to determine this?

[Response]

The protein sequences of D14 are extracted from those analyzed in Conn *et al.*, 2015, including 53 sequences from both parasitic and non-parasitic plants. 52 out of 53 D14 sequences have a V143 (number referring to ShD14). A part of the sequence alignment is added as Supplementary Fig. 9.

[Comment 22]

(line 198) GR24-5DS has a natural 2'R stereochemical configuration, but is not natural itself. This could be clearer.

[Response]

Our expression was misleading in the former version of manuscript. Therefore, the sentence has been revised as follows (lines 225-226):

"Notably, AtHTL hydrolyzes non-natural SLs with a 2'S configuration, such as GR24^{ent-5DS}, instead of GR24^{5DS}, which is of natural configuration similar to 5-deoxystrigol³³."

[Comment 23]

(line 199-200) Refer to the evidence that GR24-ent5DS is likely to avoid the steric clash.

[Response]

We have added the figure of modelled GR24^{ent-5DS} molecule in the pocket of AtHTL in Supplementary Fig. 10, showing that GR24^{ent-5DS} is likely to avoid the steric clash.

[Comment 24]

Please specify how the amino acid numbers are decided. There is some variation in the length of D14 and HTL proteins, so it would be useful to know if the numbers refer to residues in a specific protein sequence or a D14 or HTL consensus, etc. This should make it easier to compare to residues highlighted in prior studies.

[Response]

Most ShHTLs (except ShHTL2 and ShHTL8) share the same residue numbers. We used this residue number in the former version of manuscript. We have changed the residue number in the manuscript so that the number is specified for each specific protein in the revised manuscript.

[Comment 25]

(line 217-222) There is an argument that can be made that D14 in parasites likely functions similarly to D14 in other angiosperms (strong sequence conservation, maintenance of a single copy, parsimony, inability of AtD14 to rescue *kai2* germination), but I don't think that ShD14-ShMAX2 interaction or unchanged expression of ShD14 during germination are evidence that it doesn't function in germination. I would rephrase the logic here.

[Response]

We have rewritten the related statement about ShD14 as follows (lines 249-259):

"ShD14 has high sequence conservation with other angiosperms D14 (e.g., sequence identity of 75% with AtD14 and OsD14) and it maintains one *D14* copy in *Striga* as in other nonparasitic plants¹³, which strongly suggest that ShD14 has conserved function with D14 from other angiosperms. On the other hand, *AtD14* failed to rescue the germination ability of *htl* mutants¹³, suggesting that D14 is not involved in germination. Moreover, the transcripts of ShD14 remained unchanged during the conditioning and germination process of *Striga* seeds²⁷; therefore, it is likely that ShD14 does not function in germination. Instead, ShD14 might perceive endogenous SL as a phytohormone in *Striga*, similar to other D14 proteins in non-parasitic plants¹⁷, which would be consistent with the existence of endogenous SL synthesis components in *Striga*³⁵."

[Comment 26]

(Figure 5) CBB stain isn't defined or noted in the methods, although I'm guessing it's Coomassie Brilliant Blue staining. Please clarify.

[Response]

It is Coomassie Brilliant Blue staining. The full name is written in the legend for Figure 5.

[Comment 27]

(line 242-243) As the authors just pointed out in (lines 197-199), AtHTL responds to a GR24 molecule with an unnatural stereochemical configuration. Therefore, AtHTL is not mediating a SL response, although it does mediate a GR24 response.

[Response]

We have revised the related statement (lines 291-292). We also revised the word of "SL" with inappropriate use elsewhere.

[Comment 28]

(Figure 5) The Western blot data for AtHTL-AtMAX2 in Supplementary Figure 8 support a *rac*-GR24-induced interaction in pulldown. In light of this, it looks like the Y2H data in Figure 5a might show better growth for AtHTL-AtMAX2 in the presence of *rac*-GR24 than the control. It would be worth trying this particular assay again with a dilution series and replicate colonies to see if this is real and consistent with the Western results. LacZ staining may also be useful as a complementary reporter to growth, similar to Toh et al., 2014, Chemistry and Biology.

[Response]

We have performed the Y2H assays with a series of dilutions (1, 1/10, 1/20, 1/50, 1/100, 1/1000) and the results are shown in Supplementary Figure 12. The presence of *rac*-GR24 could induce better growth for both AtMAX2-AtHTL and ShMAX2-AtHTL than control and KAR₁. By contrast, in the case of ShMAX2-ShHTL1 and AtMAX2-ShHTL1 interactions, the presence of KAR₁ or *rac*-GR24 did not show difference from the control. These results seem to be consistent with those of pulldown assays. It is suggested that ShHTL1 interacts with ShMAX2 or AtMAX2 nonspecifically while the interactions between AtHTL and ShMAX2 or AtMAX2 were induced in the presence of *rac*-GR24. This is consistent with the hydrolytic activity of AtHTL towards GR24^{ent-5DS}. The related description has been added in the revised manuscript (lines 278-286).

[Comment 29]

(line 249-253, 260-264) The structural basis for high sensitivity of ShHTL7 has not been revealed, although the authors have certainly come up with some interesting hypotheses for that basis. Likewise, the basis for KAR or SL specificity or ShHTL3 specificity have not been determined, although some factors that may influence that have been identified. The significance of F150 for sensitivity has not been directly tested either. Essentially, these conclusions need to be softened as they are still untested hypotheses.

[Response]

We have revised the manuscript according to your suggestion and deleted the related expression "revealing the structural basis for its (ShHTL7) high sensitivity". Another statement about ShHTL3 has also been revised into "...P136 and F190, which were proposed as the major structural bases for the ligand specificity of ShHTL3." (lines 308-309)

Reviewer #2 (Remarks to the Author):

In this manuscript, Xu et al. analyze the D14/HTL family of SL- and KAR-binding proteins in the parasitic plant *Striga hermonthica* (Sh). *Striga* causes devastating crop losses in sub-Saharan Africa. They sense host plants by their release of SL, which function among others as chemoattractants for mycorrhiza-forming symbionts. In non-parasitic plants SL are sensed by D14 receptors, and germination-stimulating KAR by D14-homologous HTL proteins. In contrast, in *Striga*, SL are sensed by D14 and a divergent clade of HTL proteins, and stimulate germination, while KAR are bound by a smaller subclade of HTL proteins more closely related to ATHTL binds, but fail to induce germination. By employing a combination of compound- and protein-protein interaction assays with ShHTL/ShD14 structural analysis, the authors report the following major findings:

1. ShHTL1, which has been reported to be unresponsive to KAR, binds KAR with similar affinity as AtHTL and ShHTL3. Surprisingly, although all three proteins bind KAR, and the signal transducer MAX2 is required for KAR-induced germination in non-parasitic plants, KAR does not mediate HTL-MAX2 interaction. Therefore KAR-signaling shows a major difference to SL signaling, as SL function by inducing D14-MAX2 interaction.
2. The authors determined crystal the structures of ShHTL1, ShHTL4, ShHTL7, and ShD14. Structural comparison of D14 and HTL proteins from parasitic and non-parasitic plants identified pocket size, determined by the bulkiness of pocket-lining residues, as selectivity and affinity determinants. This allowed the authors to propose a “code” of a small number of aa positions that define ligand interactions and that can be utilized for the development of *Striga*-combatting “suicide germination stimulants”.

Overall, the experiments are of high quality and convincing, and of interest to a broad group of researchers studying SL/KAR signaling.

[Response]

Thank you for the careful and thorough reading of our manuscript and for pointing out the major findings of our study.

[Comment 1]

Minor points: Please correct the following overreaching statements. Lines 103-105: The result that ShHTL1 binds KAR in vitro is NOT inconsistent with the reported lack of KAR responsiveness in cross-complementation assays. Lines 157: please replace “indicate” with “suggests”. Lane 174:

The authors have NOT provided evidence that their mutants reduced SL binding. Instead they have shown that the mutants reduce SL hydrolysis, which is suggestive of reduced binding, but by no means proves it. In fact, the authors should determine binding affinities by ITC. Also, GR245DS is not a “natural” SL (it is a synthetic one). Rather, it is biologically active in inducing SL responses, whereas GR24ent-5DS is not.

[Response]

The manuscript has been revised according to the comments.

Lines 117-120:

"Our results of ShHTL1 binding KAR₁ was unexpected because it has been reported that ShHTL1 was responsive to neither KAR₁ nor *rac*-GR24 in cross-species complementation assays in terms of seed germination but rescued some other phenotypes^{13,31} (discussed later)."

Lines 338-340:

"Our observation of KAR₁ binding to ShHTL1 is unexpected because previous studies have shown that ShHTL1 was non-responsive to exogenous KAR₁ in terms of *Arabidopsis* germination and that KAR₁ failed to induce germination of *Striga*^{12,13}."

Line 177:

The word "indicated" has been replaced with "suggested".

Line 217:

It is true that only hydrolytic activity towards YLG has been examined. No binding affinity of GR24/SL can be detected by ITC when there is no heat detected. However, determination of binding affinity of GR24/SL by ITC is not appropriate because ShHTL7 is an enzyme that hydrolyzes GR24/SL after binding. The thermal changes detected by ITC are complicated because of both binding heat and reaction heat. With regard to enzyme, K_m is more appropriate in assessing substrate binding. Here we performed YLG competition assays using *rac*-GR24 and calculated the inhibition constant $K_i^{rac-GR24}$ (the reciprocal of the binding affinity of the inhibitor to the enzyme), which is more appropriate for representing the binding affinity of *rac*-GR24 to the enzyme. According to our results, $K_i^{rac-GR24}$ of ShHTL7 is smaller than ShHTL7^{L190F/C194F} and ShHTL7^{L124F/L190F/C194F}, indicating that these mutations have reduced *rac*-GR24 binding affinity. We have revised Supplementary Fig. 7 and revised the manuscript (lines 203-210).

We have also revised the following sentence accordingly (lines 225-226):

"Notably, AtHTL hydrolyzes non-natural SLs with a 2'S configuration, such as GR24^{ent-5DS}, instead of GR24^{5DS}, which is of natural configuration similar to 5DS³³."

GR24^{ent-5DS} is not inducing SL responses, but still is active in inducing some responses similar to SL responses (inhibition of *Arabidopsis* hypocotyl elongation and germination by Scaffidi A. *et al.*, Plant Physiology, 2014). GR24^{ent-5DS} is also active through AtHTL instead of AtD14. The following sentence on the activities of GR24^{5DS} and GR24^{ent-5DS} have been added (lines 226-229): "A previous study also showed that the non-natural *ent*-5DS and GR24^{ent-5DS} are able to induce responses similar to SL responses, such as inhibition of hypocotyl elongation and germination of *Arabidopsis* through AtHTL instead of AtD14³⁴."

Reviewer #3 (Remarks to the Author):

The discovery that genomes of parasitic plants of the Orobanchaceae encode a family of HTL/KAI2 proteins that can function in strigolactone (SL) and karrikin (KAR) signalling when transferred into *Arabidopsis* has stimulated much research on the structure and function of these proteins. Their role in the perception of host plant strigolactones by seeds of parasitic plants is a very important area of plant biology.

The HTL genes fall into three clades: Conserved (HTL1), Intermediate (HTL2 and 3) and Divergent (HTL4-11). There is also one putative D14 gene reported in *Striga hermonthica* which is expected to encode the putative strigolactone receptor involved in host plant development.

This new manuscript of Xu et al. evaluates KAR and SL binding activity of three *Striga* HTLs (ShHTLs) and also ShD14.

These authors have previously published data on HTL3 from the Intermediate group and provided evidence that it recognises karrikin, consistent with the karrikin response of transgenic *Arabidopsis* expressing HTL3.

The authors now show that HTL1 (Conserved clade) could bind karrikin in Isothermal Calorimetry (ITC) assays even though previous reports indicate that it did not apparently respond to karrikin in transgenic *Arabidopsis*. Xu et al now also show that HTL1 from *Arabidopsis* binds to karrikin in ITC assays.

Xu et al show that HTL4 and 7 (Divergent clade) did not bind karrikin. This is consistent with lack of karrikin response in transgenic *Arabidopsis* (previous reports).

They further showed that ShHTL1 does not hydrolyse the synthetic strigolactone GR24, whereas HTL4, 7 and 14 do. These results are consistent with data from previously published research indicating that HTLs in the Intermediate clade probably recognise strigolactones.

The authors determined crystal structures of HTL1, 4 and 7, plus ShD14 in an attempt to explain responses of these proteins to different ligands.

They further showed that HTL1 does not bind F-box protein ShMAX2. Similar results are reported for ShHTL3.

In contrast D14, HTL4 and HTL7 hydrolyse rac-GR24 slowly, and bind to ShMAX2. These results are largely consistent with other published results.

[Comment 1]

The research makes a useful contribution to the field of study but does not provide any major new insight.

[Response]

Thank you so much for your thorough reading of our manuscript. We believe that the following results of the present manuscript is new for this filed, most of which have been summarized by Reviewer #2. 1. Although the signal transducer MAX2 is believed to be required for KAR-induced germination in *Arabidopsis*, KAR does not induce HTL-MAX2 interaction as SL in inducing D14-MAX2 interaction (Toh *et al.*, 2014). Although a previous report showed that AtHTL interacted with AtMAX2 in a KAR-inducing manner, our results of Y2H (in a dilution series, Supplementary Figure 12) and pulldown assays showed that AtHTL-AtMAX2 interaction in the presence of KAR was rather weak that can be considered as nonspecific interaction. 2. We have used the recombinant proteins to detect directly the KAR-binding or GR24-hydrolyzing activities of ShHTLs and showed their different ligand specificities. This is important because *in vivo* tests are under complicated environment with more factors involved. *In vitro* examination is helpful to reinforce the results of *in vivo* tests. 3. We also solved the crystal structures of these proteins. Although structure modeling has been reported by Conn *et al.*, 2015, the smaller pocket size of ShHTL3 compared to ShHTL1 was not predicted. Therefore, our crystal structures could provide valuable information for structural bases of their different ligand specificities and for designing *Striga* germination stimulants. 4. We have shown that ShD14 is also active in hydrolyzing GR24 for the first time, which provides further information of convergent evolution of HTL and D14 in *Striga hermonthica*.

MAIN POINTS

[Comment 2]

1. A limitation of this research is that the endogenous or natural ligands for ShHTL proteins are not known. This is equally true of the D14 clade which respond to endogenous strigolactones, since the active ligands have not been identified, even in *Arabidopsis*.

[Response]

We agree that there is a limitation of our research because endogenous or natural ligands for ShHTL proteins of conserved and intermediate clades are unknown. The search of endogenous ligands of AtHTL (due to the high conservation of ShHTL1 with AtHTL, the endogenous ligands might be conserved) is of great interest and should be addressed as a future work in this research field. On the other hand, it is believed that AtHTL is mediating both KAR signaling and endogenous ligands signaling. Two different ligands for one receptor is largely because the similar structures of these two ligands. Therefore, we used the known KAR for assessment instead.

Similarly, the endogenous ligands of D14 in *Striga* or *Arabidopsis* have not been reported because endogenous SLs of known types have not been detected. It is possible that a highly unstable SL of a new type is produced in *Striga* or *Arabidopsis* or the level of endogenous SL is too low to be detected. However, *atd14* mutant showed similar phenotypes with SL synthesis mutant *max4* and *max4* mutant can be rescued by exogenous SL (Bennett *et al.*, *Biol. Open* 2016), suggesting that the product of MAX4 is highly similar to SL and might be mediated by AtD14 to induce SL response. Therefore, our finding about ShD14 hydrolyzing GR24/SL and the conservation between these D14 proteins would shed light on the function of ShD14. Nonetheless, further study would be required to identify the endogenous ligands.

[Comment 3]

While the use of a racemic mixture of GR24 isomers is convenient, and is commonly employed, it could be misleading, when aiming to elucidate the ligand binding specificity of HTL proteins. It would be far better to use stereospecific natural strigolactones such as strigol or orobanchol instead of synthetic analogues.

[Response]

We have tested the hydrolytic activity of ShHTLs towards *rac*-GR24 and natural SL 5-deoxystrigol (5DS) and the results is shown in Figure 1b. We obtained similar results between *rac*-GR24 and 5DS. It is shown that ShHTL4, ShHTL7 and ShD14 have significant hydrolytic activity towards both *rac*-GR24 and 5DS. ShHTL1 and ShHTL3 were showing no hydrolytic activity towards *rac*-GR24 while ShHTL1 had weak 5DS hydrolytic activity. In consistent, germination rate of transgenic lines transformed with different genes (*AtHTL* or *ShHTLs*) did not show significant difference between *rac*-GR24 and natural SLs such as 5DS and strigol (Toh *et al.*, *Science* 2015). We have revised the manuscript by adding the results of 5DS (lines 124-140).

[Comment 4]

2. There is increasing evidence that karrikins are not active ligands, but may be converted into active compounds in planta. Xu et al., acknowledge this. For example two studies of crystal structures of AtHTL/AtKAI2 protein with KAR1 present data showing KAR1 in different positions in the active site pocket, neither of which is closely associated with the active site amino acid residues comprising the catalytic triad. Furthermore, studies using Differential Scanning Fluorimetry (DSF) can readily detect interaction of (+)-GR24 with AtD14 and interaction of (-)-GR24 with AtHTL/AtKAI2, but no interaction with KAR (ref 35). Such results cast doubt on the validity of using KAR to study interactions with HTL proteins.

[Response]

We agree that there is possibility that KAR is converted into active compounds in planta. However, there is no direct evidence showing that KAR is not an active ligand. KAR₁ molecules observed in two crystal structures of HTL proteins were distant from the catalytic triad residues. It is likely that relocation of KAR is necessary.

DSF is useful and widely used for detecting ligand binding. But it may give rise to false negatives when there is ligand binding that does not induce any change to the thermal stability of the receptor or when the binding is so weak that the change in T_m is smaller than the chosen threshold (e.g. 0.5 °C). In the case of HTL binding KAR, hydrolysis or dramatic structural change like D14 hydrolyzing SL may not occur. By contrast, KAR binding was detected by ITC, suggesting the possibility that KAR is the ligand for HTL. Therefore, using KAR to study the KAR/GR24 specificity is valuable to distinguish the HTL-type or D14-type proteins.

[Comment 5]

In order to observe interaction the ITC analysis presented here is performed with 3mM KAR1 at a 20-fold molar excess of KAR1 relative to protein. Values for K_d are 77 uM for ShHTL1, 78 uM for ShHTL3 (previous publication), and 129 uM for AtHTL1. These values are very high. KAR1 is typically active at 1 uM in Arabidopsis, or at much lower concentrations with some karrikin-responsive species. For comparison, a K_d value of 0.3 uM is reported for rac-GR24 binding to AtD14 (ref 15).

[Response]

In fact, KAR₁ is 4-fold molar excess relative to protein, while 2-4 fold is usually used for ITC

analysis to obtain a sigmoidal curve. 202 μL of 150 μM protein sample is loaded in the sample cell and 2 μL (for each titration) of 3 mM ligand is titrated from the syringe into the sample cell. 40 μL of ligand is titrated in total, meaning that the ligand volume is about 5-fold less than the cell volume and the ligand is 4-fold molar excess relative to protein.

It is true that the K_d values for KAR binding are higher than the physiological concentration. First of all, the fraction of active protein in the sample might be lower than 100%, resulting in lower estimated affinity than real affinity. On the other hand, similar results have been observed for some abscisic acid (ABA) receptors. For instance, PYR1 has K_d for ABA of 97 μM . Surprisingly, when PYR1 forms a complex with co-receptor PP2C, binding affinities for ABA is in nanomolar range. In light of this, HTL might need other proteins or other factors to enhance KAR binding affinity. K_d of *rac*-GR24 binding to AtD14 detected by ITC is low because *rac*-GR24 binds to AtD14 and forms covalent bonds. Rather, binding along with hydrolysis occurs, making the thermal change detected by ITC too complicated to obtain K_d . The discussion about this has been added in the revised manuscript (lines 341-348).

[Comment 6]

The authors acknowledge (P5 line 16-21) that ShHTL1 expressed in Arabidopsis does not respond to karrikins. They suggest that it is possible that the heterologous protein is not expressed adequately, or another explanation is that the heterologous protein does not interact functionally with protein partners required for signal transduction. While possible, there are two arguments against these explanations; Firstly, seeds of *S. hermonthica* do not respond to KAR1. Secondly a KAI2/HTL-type protein from Selaginella can complement the Arabidopsis *kai2-1* mutant but does not respond to karrikins, GR24, debranones or carlactone (ref 35).

[Response]

As you pointed out, there are some other possibilities about this results. Actually, we have discussed this in the Discussion section (lines 325-360) including the arguments raised by the reviewer. Our former statement (line 120) might seem like there is only one possibility. Therefore, we have deleted the related discussion in the Results section.

[Comment 7]

3. Throughout the manuscript the authors should specify that they use *rac*-GR24 (not simply GR24). The different stereoisomers of GR24 have different biological activities in Arabidopsis

and behave differently in hydrolysis and binding assays.

[Response]

We have revised the manuscript using "*rac*-GR24" throughout the manuscript.

[Comment 8]

4. The authors should not assume that GR24 and natural strigolactones behave the same, and so should be extremely careful not to equate GR24 with 'SL'. For example P6 lines 11-12.

[Response]

We have been cautious about these terms in the revised manuscript.

MINOR POINTS

[Comment 9]

Abstract. 2-3. There is little evidence that SL is a germination stimulant in *Arabidopsis* and none in rice.

[Response]

We agree with the reviewer. Our expression has been revised as "HYPOSENSITIVE TO LIGHT (HTL) mediates the perception of both karrikin and the germination stimulant of the parasitic weed *Striga*, strigolactone. Strigolactone is also a phytohormone that needs DWARF14 (D14) as a receptor." (lines 22-24).

[Comment 10]

Intro p3 line 14. Use of the word 'endogenous' is misleading because the endogenous SL is unknown and there is no endogenous karrikin.

[Response]

We have deleted the word "endogenous".

[Comment 11]

16-18. Although it is correct to say that D14 requires MAX2 as a signalling component, this is not so for HTL/KAI2 because there is little evidence for interaction between HTL/KAI2 and MAX2 (as this manuscript confirms). Lines 20-24 are accurate. Lines 16-18 should be modified.

[Response]

We have rewritten the related statement as follows (lines 52-54):

"D14 require the F-box protein MAX2 (MORE AXILLARY GROWTH2) as the SL signal transducing component and MAX2 is also required for KAR signaling²¹⁻²⁵."

[Comment 12]

18-20. The attack on SL by D14 is not truly hydrolysis because the final step is not completed.

[Response]

We have replaced the former phrase "D14 hydrolyzes SL ..." with "D14 is attacked by SL ..." (line 54).

[Comment 13]

P4 line 2. 'Perception in roots' is confusing. Perception by seeds?

[Response]

Our former expression is not correct and it should be "perception of exogenous SL by seeds". We have revised the corresponding words (line 62).

[Comment 14]

Line 8 (and elsewhere throughout the manuscript). Genes are not transformed into plants; Plants are transformed with genes.

[Response]

We have revised the related expression as follows:

"*Arabidopsis htl* mutants were transformed with *ShHTLs* genes" (lines 73)

"The SL and/or KAR responsiveness of *ShHTLs* have been characterized using transgenic *Arabidopsis* lines transformed with *ShHTL* genes in the *htl* mutant background^{13,28}." (lines 103-105)

[Comment 15]

9-10. The identity of SLs should be given since results can vary depending on which compounds are used.

[Response]

According to the previous papers (Toh *et al.*, 2015 and Conn *et al.*, 2015), we have added the following description (lines 70-72):

"(including natural SLs: strigol, 5-deoxystrigol (5DS), 2'-*epi*-5-deoxy strigol and sorgolactone and synthetic SL: *rac*-GR24)"

Although EC₅₀ values of these natural or synthetic SLs are slightly different, these SLs have the same activities (in terms of germination).

[Comment 16]

12. Define YLG

[Response]

As pointed out by the reviewers, our former description of YLG as a fluorescent compound is not correct because it does not become fluorescent until it is hydrolyzed. Therefore, definition of YLG has been added as follows (lines 76-78):

"a profluorogenic SL agonist that becomes fluorescent when it is hydrolyzed by D14 or some HTL enzymes".

[Comment 17]

15-23. This paragraph is a summary of results, but should explain the aims. Lines 17-18 and 21-23 are cryptic, not informative.

[Response]

We agree with your comments and our aims have been stated more clearly in the revised manuscript as follows:

"Despite that these ShHTLs proteins have high sequence conservation (sequence identity of more than 60%), they have distinct *rac*-GR24/KAR specificities and become great candidates for the

study of SL/KAR perception. Unveiling the structural basis for the ligand specificities of this series of proteins will help us understand the evolution and discrimination of SL and KAR signals." (lines 86-91)

"However, it remains unclear whether ShHTLs of different clades are able to interact with ShMAX2 to transduce signals like D14-MAX2 in *Arabidopsis*. Therefore, interactions between other members of ShHTLs and ShMAX2 were determined comprehensively in the present study."
" (lines 96-99)

[Comment 18]

P5. Line 3-4. Re-word and explain fully. Mutants are organisms resulting from mutations.

[Response]

As a revision from another comment, we have revised the wrong expression as "genes transforming into plants". Since Conn *et al.*, 2015 and Toh *et al.*, 2015 have used the same expression "in the *htl* mutant background", we would like to use "mutant" here to mean the organism of *htl* mutant. Therefore, the related sentence has been revised as follows (lines 103-105):

"The SL and/or KAR responsiveness of ShHTLs have been characterized using transgenic *Arabidopsis* lines transformed with *ShHTL* genes in the *htl* mutant background^{13,28}."

[Comment 19]

P6 line 8 'hydrolysis assays'

[Response]

We have replaced "hydrolyzing assays" with "hydrolysis assays" (line 132).

[Comment 20]

Line 9. 'Data not shown'. Data should be shown.

[Response]

We have added the data showing the YLG hydrolysis by ShD14 and OsD14 in Supplementary Fig. 4 as you suggested.

[Comment 21]

Line 10-11. 'It has been suggested that ShD14 also serves as an SL receptor.' Provide reference.

[Response]

We intended to say "It is suggested by this study that..." In fact, there is no report about ShD14 except Tsuchiya *et al.*, 2015, which examined the hydrolytic activity of YLG only. Therefore, the related sentence has been revised as follows (lines 134-135):

"It is suggested that ShD14 also serves as an SL receptor."

[Comment 22]

Line 12. 'hydrolytic activity towards SL. ShHTL4, ShHTL7 and ShD14 were able to hydrolyze SL' Authors should be specific and refer to rac-GR24 instead of SL, since no natural SLs have been investigated.

[Response]

We agree with the comment that we should be specific about which SL/GR24 is used. We have revised this paragraph and elsewhere throughout the manuscript.

[Comment 23]

P8. Lines 10. It is confusing to refer to double and triple mutants because only a single gene is mutated. Better to refer to mutations which produce HTL protein with two or three and amino acid substitutions.

[Response]

We agree that it is better to refer to mutation when only a single gene is mutated. Therefore, "double mutation" and "triple mutation" have been used in the revised manuscript.

[Comment 24]

P11. Lines 7-9. 'Since AtHTL is able to hydrolyze GR24 (30,31,35) and is involved in GR24-inducing germination (13), the results of our pull-down assays support the hypothesis that AtHTL mediates SL response, to some extent, via the interaction with AtMAX2.' I think this interpretation is highly questionable because the published genetic evidence is that SL does not

induce signalling via AtHTL. One stereoisomer of GR24 is active. While *Atmax2* mutants do not respond to KAR, this does not provide evidence for functional interaction between MAX2 and HTL.

[Response]

We agree that SL does not induce signaling via AtHTL. Instead, AtHTL mediates a non-natural GR24 response and regulates plant growth such as inhibition of hypocotyl elongation similar to SL response. Therefore, our data suggest that AtHTL mediates the GR24 response similar to D14 mediating SL response. On the other hand, previous reports showed that AtHTL hydrolyzes GR24^{ent-5DS} and is thermally unstable in the presence of GR24^{ent-5DS}, suggesting that structural changes might occur, which is necessary for binding MAX2 like D14 does.

We have revised the manuscript as follows:

"Since AtHTL is able to hydrolyze GR24^{ent-5DS}, becomes thermally unstable in the presence of GR24^{ent-5DS} similar to D14 upon binding SL and AtHTL mediates a GR24^{ent-5DS} response to regulate plant growth such as inhibition of hypocotyl elongation similar to SL response, it is likely that AtHTL mediates the GR24^{ent-5DS} response in a similar pathway to SL response mediated by D14. Therefore, the results of our pull-down assays support the hypothesis that, to some extent, AtHTL mediates the GR24^{ent-5DS} response via the interaction with AtMAX2." (lines 286-293)

No response to KAR of *atmax2* mutants does not indicate there is functional interaction between MAX2 and HTL. As suggested by your next comment, it might be pleiotropic. However, the high sequence conservation and structure similarity between D14 and HTL might suggest that they have a similar partner or a similar signaling pathway. Therefore, it is worth trying to test whether there is interaction between HTL and MAX2. There are a lot of possibilities as discussed (lines 325-360).

[Comment 25]

P12, line 9. We need to be more cautious about MAX2 being 'involved' in KAR response since *max2* mutants might simply be pleiotropic.

[Response]

The related sentence has been changed to "MAX2 is required for KAR signaling" (line 316).

[Comment 26]

Overall the Discussion should be more cautious about the use of unnatural chemicals (KAR and *rac*-GR24) to study ligand specificity of the HTL and D14 proteins.

[Response]

We have been more cautious about the use of KAR and SL throughout the manuscript. KAR₁ and *rac*-GR24 have been used in the revised manuscript.

REVIEWERS' COMMENTS:

Reviewer #1 (Remarks to the Author):

I am satisfied with the authors' responses to my comments.

Reviewer #2 (Remarks to the Author):

The authors have satisfactorily responded to my requests, and the manuscript is significantly improved in response to requests from all three reviewers. I recommend acceptance of the manuscript.

Minor comments: The language of the newly added sections should be improved, which can be done at the editing stage. For the new Fig. 1b, please provide statistical analysis for the claim that "ShHTL1 and ShHTL3 were showing no hydrolytic activity towards rac-GR24 while ShHTL1 had weak 5DS hydrolytic activity".

Reviewer #3 (Remarks to the Author):

The authors have attended to most of the points raised by reviewers. The manuscript has undergone numerous changes and requires careful editing by a scientific writer to improve precision and clarity. Some comments and editing have been made and on the attached pdf, but further editing is strongly recommended.

A new paper by Yao et al. (Plant J. 2018), is highly relevant and should be cited.

The authors should be careful to distinguish *htl1* and *kai2* mutants since they have different genetic backgrounds and potentially have different responses or phenotypes. When citing a particular paper or experiment, the correct mutant (either *htl1* or *kai2*) should be referred to. I believe that YLG is a fluorogenic compound, not profluorogenic.

[Editorial Note: Due to the journal's embargo policy, an annotated manuscript file cannot be reproduced within this Peer Review File.]

We thank the reviewers for their review of our manuscript. The following are point-by-point responses to the reviewers' comments on our manuscript entitled "Structural analysis of HTL and D14 proteins reveals the basis for ligand selectivity in *Striga*".

REVIEWERS' COMMENTS:

Reviewer #1 (Remarks to the Author):

[Comment 1]

I am satisfied with the authors' responses to my comments.

[Response]

Thank you for your review of our manuscript.

Reviewer #2 (Remarks to the Author):

[Comment 2]

The authors have satisfactorily responded to my requests, and the manuscript is significantly improved in response to requests from all three reviewers. I recommend acceptance of the manuscript.

Minor comments: The language of the newly added sections should be improved, which can be done at the editing stage. For the new Fig. 1b, please provide statistical analysis for the claim that "ShHTL1 and ShHTL3 were showing no hydrolytic activity towards *rac*-GR24 while ShHTL1 had weak 5DS hydrolytic activity".

[Response]

Thank you for your suggestions. We have revised our manuscript using a professional English editing service.

We have performed statistical analysis for the data in Fig. 1b. In the case of *rac*-GR24, ShHTL1 and ShHTL3 are showing no significant hydrolytic activity. In the case of 5DS, ShHTL3 has no significant hydrolytic activity while ShHTL1 has significant hydrolytic activity compared to buffer. However, the activity of ShHTL1 is far weaker than those of ShHTL4, ShHTL7 and ShD14. The statistical analysis is shown in the revised Fig. 1b.

Reviewer #3 (Remarks to the Author):

[Comment 3]

The authors have attended to most of the points raised by reviewers. The manuscript has undergone numerous changes and requires careful editing by a scientific writer to improve precision and clarity.

Some comments and editing have been made and on the attached pdf, but further editing is strongly recommended.

[Response]

Thank you for your valuable comments. All your comments have been addressed. We have also revised our manuscript using a professional English editing service.

Page 3 Line 58:

“HTL” has been changed to “HTL/KAI2”.

Page 3 Line 59 - Page 4 Line 61:

A recent report shows that AtKAI2 exhibits auto-activation artefacts in Y2H experiments (Yao *et al.*, *Plant J.* 2018). We have confirmed that auto-activation artefacts are inhibited in our system as shown in Supplementary Fig. 13. In fact, Yao *et al.* used a different Y2H system (i.e. different reporter genes and selective medium) from us.

Page 4 Line 64:

“*Striga*” has been changed to “*Striga hermonthica*”.

Page 4 Line 65:

“Eleven ShHTLs (*S. hermonthica* HTLs)” has been changed to “Eleven *S. hermonthica* HTLs (ShHTLs)”.

Page 4 Lines 71-75:

The sentence have been revised as

“ShHTLs of the different clades showed distinct responses to SL, including natural SLs: strigol, 5-deoxystrigol (5DS), 4-deoxyorobanchol and sorgolactone, and a racemic mixture of the synthetic SL analog GR24 (*rac*-GR24) or KAR in cross-species complementation assays, in which *Arabidopsis kai2-2* and *htl-3* mutants were transformed with *ShHTLs* genes...”

Page 4 Lines 77-79:

The sentence have been revised as

“Despite these ShHTLs proteins having high sequence conservation (sequence identity of more than 60%), they have distinct SL/KAR specificities and are therefore important for the study of SL/KAR perception.”

Page 4 Lines 83-84:

We agree with the reviewer and we have changed “profluorogenic” to “fluorogenic” in the revised manuscript.

Since only AtD14 has been reported to be able to hydrolyze YLG, “D14” here has been changed to “AtD14”.

Page 5 Line 106:

“determined by” has been changed to “determined”.

Page 6 Lines 121-122:

“detecting the amount of SL after enzyme treatment using HPLC” has been changed to “detecting the amount of SL by high-performance liquid chromatography (HPLC) after enzyme treatment”

Page 6 Line 125:

“Tsuchiya, Y. et al²⁸” has been changed to “previously published²⁸”

Page 6 Lines 129-131:

The sentence has been reworded:

ShD14 was able to hydrolyze *rac*-GR24 and 5DS but not YLG, suggesting that ShD14 might also serve as an SL receptor.

Page 7 Lines 135-137:

“without KAR₁-binding ability” has been changed to “but lacked KAR₁-binding ability”.

“In contrast” has been added to the head of the following sentence “ShHTL1 and ShHTL3 are KAR₁-binding proteins without hydrolytic activity towards *rac*-GR24.”

Page 8 Line 156:

“The 150th residue” has been replaced by “residue 150”. And this format has been used throughout the revised manuscript.

Page 10 Line 216:

“Binding/hydrolyzing ability” is replaced by “binding and hydrolyzing ability”

Page 10 Line 226:

The apostrophe in “2' S configuration” has been changed to prime “2' S configuration”. The whole manuscript has been revised.

Page 15 Lines 327-328:

The sentence “and the interaction between AtHTL and AtMAX2 has been questioned²⁷” has been added.

[Comment 4]

A new paper by Yao et al. (Plant J. 2018), is highly relevant and should be cited.

[Response]

We have cited the new paper by Yao *et al.* (Plant J. 2018) as Ref. 27 and relevant contents are discussed in the revised manuscript. (Page 3 Line 59-Page 4 Line 61; Page 4 Line 81- Page 5 Line 85; Page 9 Lines 198-201 and Page 13 Lines 284-287).

[Comment 5]

The authors should be careful to distinguish *htl1* and *kai2* mutants since they have different genetic backgrounds and potentially have different responses or phenotypes. When citing a particular paper or experiment, the correct mutant (either *htl1* or *kai2*) should be referred to.

[Response]

We have not referred to *htl1* mutant in the manuscript. We supposed that the reviewer was meaning *htl* mutant should be distinct from *kai2* mutant. In the revised manuscript, we used *kai2-2* (Waters *et al.* 2012; Conn *et al.* 2015; Conn *et al.* 2016 and Waters *et al.* 2014) and *htl-3* mutants (Toh *et al.* 2015) instead to indicate the specific mutant used in a particular paper (Page 3 Line 59; Page 4 Line 74; Page 5 Line 99; Page 12 Line 254 and Page 16 Line 354).

[Comment 6]

I believe that YLG is a fluorogenic compound, not profluorogenic.

[Response]

We agree with the reviewer and we have changed “profluorogenic” to “fluorogenic” in the revised manuscript (Page 4 Line 83).